# Centralized Reward Agent for Knowledge Sharing and Transfer in Multi-Task Reinforcement Learning

**Haozhe Ma**[1,2], **Zhengding Luo**[3],[*] **Thanh Vinh Vo**[1], **Kuankuan Sima**[4], **Tze-Yun Leong**[1]

[1]School of Computing, National University of Singapore
[2]TikTok Pte. Ltd., Singapore
[3]School of Electrical and Electronic Engineering, Nanyang Technological University
[4]Department of Electrical and Computer Engineering, National University of Singapore
{haozhe.ma, kuankuan_sima}@u.nus.edu,
{votv, leongty}@nus.edu.sg, luoz0021@e.ntu.edu.sg

## Abstract

Reward shaping is effective in addressing the sparse-reward challenge in reinforcement learning (RL) by providing immediate feedback through auxiliary, informative rewards. Based on the reward shaping strategy, we propose a novel multi-task reinforcement learning framework that integrates a centralized reward agent (CRA) and multiple distributed policy agents. The CRA functions as a knowledge pool, aimed at distilling knowledge from various tasks and distributing it to individual policy agents to improve learning efficiency. Specifically, the shaped rewards serve as a straightforward metric for encoding knowledge. This framework not only enhances knowledge sharing across established tasks but also adapts to new tasks by transferring meaningful reward signals. We validate the proposed method on both discrete and continuous domains, including the representative Meta-World benchmark, demonstrating its robustness in multi-task sparse-reward settings and its effective transferability to unseen tasks.

## 1 Introduction

Reinforcement learning (RL) has made significant progress across various domains, such as robotics [Kober et al., 2013], gaming [Lample and Chaplot, 2017], autonomous vehicles [Aradi, 2020], signal processing [Luo et al., 2024], and large language models [Shinn et al., 2023, Ouyang et al., 2022]. However, environments with sparse and delayed rewards remain a significant challenge, as the absence of immediate feedback hinders the agent from distinguishing the value of states and leads to aimless exploration [Ladosz et al., 2022]. Reward Shaping (RS) has been proven to be an effective technique for addressing this challenge by providing additional dense and informative rewards [Sorg et al., 2010b,a]. Concurrently, multi-task reinforcement learning (MTRL) is becoming increasingly important due to its ability to transfer knowledge across tasks. In this context, the auxiliary rewards infused with task-specific information in RS offer a straightforward means to distribute knowledge among different tasks. Integrating RS techniques into MTRL is a highly promising and intuitive direction to enhance the efficacy of multi-task learning systems.

Numerous MTRL algorithms for knowledge transfer have been developed. Policy distillation methods identify and combine commonalities across different policies [Rusu et al., 2016, Teh et al., 2017, Parisotto et al., 2016, Xu et al., 2024]; representation sharing methods extract and share the common features or gradients among agents [Yang et al., 2020, D'Eramo et al., 2020, Sodhani et al., 2021]; and parameter sharing methods design architectural modules to reuse parameters or layers across networks [Sun et al., 2022, Cheng et al., 2023]. Despite their potential, these strategies often face slow

---

[*]Corresponding author.

39th Conference on Neural Information Processing Systems (NeurIPS 2025).

adaptation to and limited utilization of transferred knowledge. Therefore, leveraging reward shaping, which directly adds a metric to the reward function, offers a compelling alternative to address these limitations.

Regarding reward shaping, not all shaped rewards effectively serve as a medium for knowledge transfer. Specifically, the intrinsic-motivation-based rewards are typically designed using heuristics to generate task-agnostic signals. Examples include incorporating exploration bonuses [Bellemare et al., 2016, Ostrovski et al., 2017, Devidze et al., 2022], rewarding novel states [Tang et al., 2017, Burda et al., 2018], and encouraging curiosity-driven behaviors [Pathak et al., 2017, Mavor-Parker et al., 2022]. Although these approaches encourage broader exploration, they are not directly related to specific tasks and thus lack transferability. Consequently, we focus on another branch of RS methods, task-contextual rewards, which automatically learn and encode task-specific information, such as hidden values, states contributions, or future-oriented insights, that can be effectively shared across various tasks [Ma et al., 2024a, 2025b,a, Mguni et al., 2023, Memarian et al., 2021].

To share task-related knowledge in MTRL via RS techniques, and inspired by the *ReLara* framework [Ma et al., 2024a], which integrates an assistant reward agent to densify sparse environmental rewards, we propose the **Cen**tralized Reward Agent based MTRL f**RA**mework (**CenRA**)[2]. The framework consists of two main components: a *centralized reward agent* (CRA) and multiple distributed *policy agents*. Each policy agent individually learns control behaviors within its respective tasks and shares its experiences with the CRA. The CRA extracts common knowledge from these experiences and learns to generate dense rewards that are encoded with task-specific information. These rewards are then distributed back to the policy agents to augment their original environmental rewards. Additionally, given that different tasks may contribute variably to the MTRL system, we introduce an information synchronization mechanism to further balance knowledge distribution by considering task similarity and agent learning progress, thereby ensuring system-wide optimal performance. The main contributions of this paper are summarized as follows:

- (*i*) We propose the CenRA framework to address MTRL problems. It incorporates a CRA that functions as a knowledge pool, efficiently distilling and distributing valuable information from various tasks to policy agents while adapting to new tasks.
- (*ii*) CenRA leverages reward shaping techniques to infuse insights via dense rewards. This approach not only provides a direct signal for policy agents to absorb knowledge but also effectively addresses the sparse-reward challenge.
- (*iii*) We introduce an information synchronization mechanism that considers both task similarity and agent learning progress to balance multi-task learning. This mechanism provides a novel direction for maintaining system equilibrium in MTRL.
- (*iv*) CenRA is validated in both discrete and continuous control MTRL environments with sparse extrinsic rewards. CenRA outperforms baseline models in learning efficiency, knowledge transferability, and system-wide performance.

## 2 Related Work

Multi-task reinforcement learning (MTRL) has attracted significant attention recently due to its potential to share knowledge across multiple tasks, thereby improving learning performance [Caruana, 1993]. We discuss existing MTRL literature from three main directions:

**Knowledge Transfer** methods focus on identifying and transferring task-relevant features across diverse tasks [Zeng et al., 2021]. Policy distillation [Rusu et al., 2016] is a well-studied approach to extract and share task-specific behaviors or representations that many works are built on: Teh et al. [2017] introduced *Distral*, which distills a centroid policy from multiple task-policies; Parisotto et al. [2016] developed *Actor-Mimic*, where a single policy is trained to mimic several expert policies from different tasks; while Yin and Pan [2017] incorporated hierarchical prioritized experience replay buffer to select and learn multi-task experiences; Hessel et al. [2019] further proposed an adaptation mechanism to equalize the impact of each task in policy distillation. Additionally, Xu et al. [2020] explored the transfer of offline knowledge to train policies, and further leveraged online learning for fine-tuning. Bai et al. [2023] introduced a dual-phase learning approach, optimizing individual policies while correcting them across multiple tasks. Mysore et al. [2022] used separate critics for each task to accompany a single actor to integrate their feedback. These methods mitigate gradient

---

[2]The source code is accessible at: `https://github.com/mahaozhe/CenRA`

interference to an extent, however, balancing the distribution of knowledge across tasks is crucial. Without a careful trade-off, the performance of the entire system could be compromised.

**Representation Sharing** methods explore architectural solutions of reusing network modules or representing commonalities to the MTRL problem [D'Eramo et al., 2020, Devin et al., 2017, Hong et al., 2021, Ma et al., 2024b, 2023]. Sun et al. [2022] used a parameter compositional approach to learn and share a subspace of parameters, allowing policies for various tasks to be interpolated within it. Yang et al. [2020] employed soft modularization to learn foundational policies and utilized a routing network to generate probabilities to combine them. He et al. [2024] introduced the Dynamic Depth Routing framework, which dynamically adjusts the use of network modules in response to task difficulty. Sodhani et al. [2021] leveraged task-related metadata to create composable representations. Cheng et al. [2023] and Lan et al. [2023] both incorporated attention mechanisms: the former employed attention-based mixture of experts to capture task relationships, while the latter used Temporal Attention for contrastive learning purposes. Although these methods demonstrate efficacy in learning shared representations, they may struggle to fully capture the complexity of highly diverse tasks. Moreover, adapting shared structures to new tasks typically requires extra design efforts.

**Single-Policy Generalization** methods learn a single policy to solve multiple tasks simultaneously or continuously, in the absence of information from prior policies or task-specific details, in which case, the primary goal is to enhance the policy's generalization capabilities. Model-free meta-learning techniques have been proposed to enhance the multi-task generalization [Finn et al., 2017]. Yang et al. [2017] designed a sharing network structure that allows an agent to learn multiple tasks concurrently. Vuong et al. [2019] introduced a confidence-sharing agent to detect and define shared regions between tasks to support single policy learning. Wan et al. [2020] proposed a transfer learning framework to handle mismatches in state and action spaces. Additionally, several methods focus on overcoming gradient interference to enhance the generalization in various tasks [Chen et al., 2018, Yu et al., 2020a], while Ammar et al. [2014] developed a consecutive learning policy gradient approach. These methods are efficient in saving computational resources, but the generalization ability of the policy may be constrained when faced with out-of-distribution or previously unseen tasks.

## 3 Preliminaries

**Markov Decision Process (MDP)** models sequential decision-making problems under uncertainty. An MDP represents the interaction between an agent and its environment as a tuple $\langle S, A, P, R, \gamma \rangle$, where $S$ is the state space, $A$ is the action space, $P : S \times A \times S \to [0, 1]$ is the probability of transiting from one state to another given an action, $R : S \times A \to \mathbb{R}$ is the reward function, and $\gamma \in [0, 1]$ is the discount factor to modulate the importance of future versus immediate reward.

**Multi-Task Reinforcement Learning (MTRL)** addresses the challenge of learning multiple tasks simultaneously within an integrated model to leverage commonalities and differences across tasks. Typically, MTRL introduces a task space $\mathcal{T}$, assuming all tasks are sampled from this space and thus follow a unique distribution. Each task is modeled as an independent MDP. An MTRL agent aims to learn optimal policies $\pi_i : S \to A$ for each task $T_i \sim \mathcal{T}$, to maximize their corresponding expected cumulative rewards, or returns, denoted by $G_i = \mathbb{E}[\sum_{t=0}^{\infty} \gamma^t R_i(s_t, a_t)]$.

**RL with an Assistant Reward Agent (ReLara)** [Ma et al., 2024a] introduces a dual-agent framework designed to tackle the challenge of sparse rewards in RL. Within this framework, the original agent is termed as *policy agent*, while an assistant *reward agent* is integrated to enrich the feedback mechanism by generating dense, informative rewards. The reward agent, trained as a self-contained RL agent, autonomously extracts hidden value information from the environmental states and the actions of the policy agent to craft meaningful reward signals. These signals significantly improve learning efficiency by providing immediate and pertinent bonuses.

## 4 Methodology

We propose the **Cen**tralized Reward Agent f**RA**mework (**CenRA**) for MTRL, which incorporates a *centralized reward agent* (CRA) to support multiple reinforcement learning agents across multiple tasks. A high-level illustration of the CenRA framework is shown in Figure 1. The CRA is responsible for extracting general task-specific knowledge from various tasks and distributing valuable information to the policy agents by reconstructing their reward models. The detailed methodology

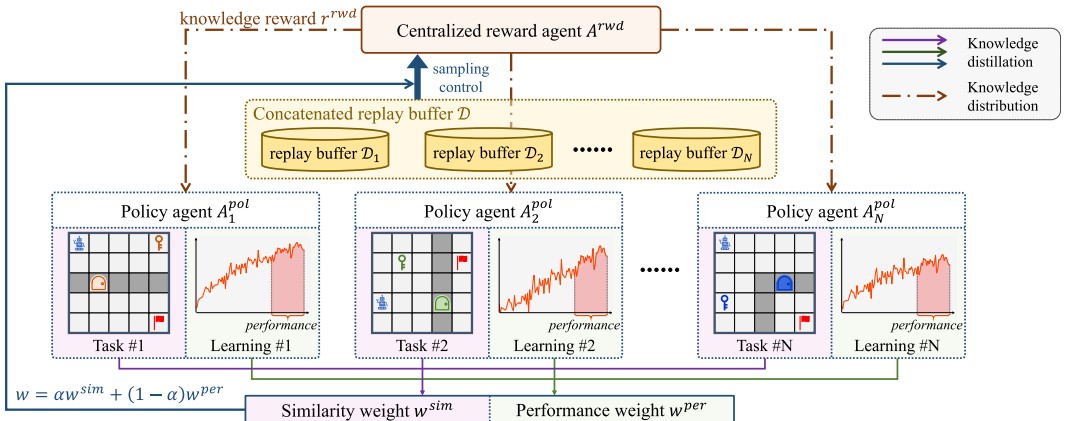

Figure 1: A high-level illustration of the CenRA framework. The centralized reward agent functions as a knowledge repository, distilling information from various tasks and distributing it to individual policy agents to enhance learning efficiency.

for knowledge extraction and sharing is presented in Section 4.1. Furthermore, to mitigate the potential disparities in the information that each task contributes, which might lead to an imbalance in knowledge distribution, we introduce an information synchronization mechanism by considering two main factors: the similarity of the tasks and the online learning performance of the policy agents, details given in Section 4.2. Finally, the overall framework of CenRA is presented in Section 4.3.

## 4.1 Knowledge Distillation and Distribution

### 4.1.1 Problem Formulation

We consider an MTRL setting comprising $N$ distinct tasks $\{T_1, T_2, \ldots, T_N\}$, all executed within the same type of environment $\mathcal{E}$. We assume that the shape of state $s \in S$ and action $a \in A$ remain uniform across tasks, to ensure the CRA processes consistent inputs. Despite this uniformity, each task may feature different state spaces, action spaces, goals, and transition dynamics. For instance, a series of mazes with the same size but varying map configurations would satisfy this condition. For each task $T_i$, we denote the transition function as $P_i(s'|s, a)$ and the reward function as $R_i(s, a)$.

The centralized reward agent (CRA) is denoted as $\mathcal{A}^{rwd}$ and multiple policy agents are denoted as $\{\mathcal{A}_1^{pol}, \mathcal{A}_2^{pol}, \ldots, \mathcal{A}_N^{pol}\}$. Each policy agent $\mathcal{A}_i^{pol}$ operates independently to complete its corresponding task $T_i$, utilizing appropriate RL algorithms as backbones. For example, implementing DQN [Mnih et al., 2015] for discrete control tasks, while TD3 [Fujimoto et al., 2018] or SAC [Haarnoja et al., 2018a] for continuous control tasks. Moreover, the policy of CRA $\mathcal{A}^{rwd}$ is $\pi^{rwd}$, and the internal policy of policy agent $\mathcal{A}_i^{pol}$ is $\pi_i^{pol}$.

### 4.1.2 Centralized Reward Agent

The CRA $\mathcal{A}^{rwd}$ aims to extract environment-relevant knowledge and distribute it to policy agents by generating additional dense rewards to support their original reward functions. Similar to the ReLara framework [Ma et al., 2024a], we model the CRA as a self-contained RL agent, yet, as an extension to ReLara, our CRA is designed to concurrently interact with multiple policy agents and their respective tasks. The CRA's policy $\pi^{rwd}$ generates continuous rewards given both an environmental state and a policy agent's behavior. Specifically, $\pi^{rwd}$ maps the Cartesian product of the state space and action space, $S \times A$, to a defined *reward space*, which constrains the rewards to a range of real numbers, $\mathcal{R} = [R_{min}, R_{max}] \subset \mathbb{R}$. For simplicity, we denote the observation of the CRA as $s^{rwd} = (s_i, a_i)$, where $s_i \sim T_i$ and $a_i \sim \pi_i^{pol}(s_i)$. To distinguish from the environmental reward, the generated reward is termed as *knowledge reward*, denoted as $r^{knw}$.

We adopt an off-policy actor-critic algorithm to optimize the CRA [Konda and Tsitsiklis, 1999]. To aggregate and reuse experiences from all policy agents, a concatenated replay buffer $\mathcal{D} = \bigcup_{i=1}^{N} \mathcal{D}_i$ is

constructed, where $\mathcal{D}_i$ represents the replay buffer of each policy agent $\mathcal{A}_i^{pol}$. Besides, each transition is augmented with the CRA-generated knowledge reward, $r^{knw}$. Specifically, the transition from policy agent $\mathcal{A}_i^{pol}$ stored in the replay buffer is defined as $\tau = (s_t^{rwd}, r_t^{knw}, r_t^{env}, s_{t+1}^{rwd}|T_i)$. The augmented transition includes all necessary information for optimizing both the CRA and each corresponding policy agent, thus making the concatenated replay buffer a shared resource across the entire framework and minimizing storage overhead.

The CRA's update process involves using these stored transitions to optimize the reward-generating actor $\pi^{rwd}$ and the value estimation critic. The objective function for the critic module is:

$$J(V^{rwd}) = \mathbb{E}_{\tau_t \sim \mathcal{D}}[\delta_t^2], \quad \delta_t = r_t^{env} + \gamma V^{rwd}(s_{t+1}^{rwd}) - V^{rwd}(s_t^{rwd})|T_i, \tag{1}$$

where $\tau_t = (s_t^{rwd}, r_t^{env}, s_{t+1}^{rwd}|T_i) \sim \mathcal{D}$. Concurrently, the actor module is updated through the following objective function:

$$J(\pi^{rwd}) = \mathbb{E}_{\tau_t \sim \mathcal{D}}\left[\mathbb{E}_{r_t^{knw} \sim \pi^{rwd}(\cdot|s_t^{rwd})}\left[\log \pi^{rwd}(r_t^{knw}|s_t^{rwd}) \cdot \delta_t\right]\right]. \tag{2}$$

### 4.1.3 Policy Agents with Knowledge Rewards

Each policy agent $\mathcal{A}_i^{pol}$ stores the experiences in its corresponding replay buffer $\mathcal{D}_i$. They receive two types of rewards: the environmental reward $r_i^{env}$ from their respective task $T_i$ and the knowledge reward $r^{knw}$ from CRA. The augmented reward is given by:

$$r_i^{pol} = r_i^{env} + \lambda r^{knw}, \quad r^{knw} \sim \pi^{rwd}(\cdot|s_i, a_i), \tag{3}$$

where $\lambda \in (0, 1]$ is a scaling weight factor. The optimal policy $\pi_i^{pol*}$ for each agent is derived by maximizing the cumulative augmented reward:

$$\pi_i^{pol*} = \arg\max_{\pi_i^{pol}} \mathbb{E}_{(s_i, a_i) \sim \pi_i^{pol}}\left[\sum_{t=0}^{\infty} \gamma^t r_i^{pol}\right]. \tag{4}$$

It is worth noting that the environmental reward $r_i^{env}$ is retrieved from the replay buffer (if adopting an off-policy approach). Conversely, the knowledge reward $r^{knw}$ is computed in real-time using the most recently updated $\mathcal{A}^{rwd}$, ensuring it reflects the latest learning advancements. Lastly, each policy agent is able to employ any suitable RL algorithm, whether on-policy or off-policy, to best address its specific task, which enhances the CenRA framework's generality and flexibility.

### 4.2 Information Synchronization of Policy Agents

In the CenRA, the information provided by different tasks may exhibit significant disparities, potentially leading to an imbalance in knowledge extraction and distribution. We introduce an information synchronization mechanism for CenRA to maintain a balanced manner from the perspective of the entire system. Specifically, we control the quantity of samples that CRA retrieves from each task's replay buffer $\mathcal{D}_i$ by a *sampling weight* $w$, by considering two aspects: the similarity among tasks and the real-time learning performance of the policy agents.

**Similarity Weight $w^{sim}$** is derived from the similarity among tasks, enabling the CRA to focus on relatively outlier tasks. To simplify computation, we use the hidden layers extracted from each policy agent's neural network encoders to represent the tasks' features. To reduce randomness, we average the hidden features of the most recent $K$ steps. We adopt a cross-attention mechanism to calculate the similarity weight [Vaswani et al., 2017]. Specifically, for task $T_i$, let $\boldsymbol{H}_i$ denote the averaged hidden feature vector, which serves as the *key*, and the centroid of all tasks $\boldsymbol{c}$ acts as the *query*. Then, the similarity $s_i$ of task $T_i$ to the centroid of the task cluster is calculated as:

$$s_i = \frac{\boldsymbol{c}^T \cdot \boldsymbol{H}_i}{\sqrt{D}}, \quad \boldsymbol{c} = \frac{1}{N}\sum_{k=1}^{N} \boldsymbol{H}_k, \tag{5}$$

where $D$ is the dimension of the hidden feature to prevent gradient vanishing or exploding. A larger $s_i$ indicates a greater similarity between $T_i$ and the centroid. It is worth noting that, to avoid the centroid $\boldsymbol{c}$ approaching zero due to feature vectors $\boldsymbol{H}_i$ pointing in opposite directions, all latent representations

$\boldsymbol{H}_i$ in our framework are extracted from ReLU activation layers. This ensures that every element of $\boldsymbol{H}_i$ is non-negative, effectively preventing feature cancellation and maintaining a well-defined and numerically stable centroid $\boldsymbol{c}$. Given our assumption is that the tasks farther from the centroid require more attention, the *similarity weight* is defined as $\boldsymbol{w}^{sim} = \text{Softmax}\big([1/s_1, 1/s_2, \ldots, 1/s_N]\big)$.

**Performance Weight** is determined by the real-time learning performance of each policy agent, to ensure the CRA focuses more on lagging tasks. Similar to the similarity weight, we average the environmental rewards $r_i^{env}$ from the most recent $K$ steps, denoted as $R_i^{tail}$, to measure the recent learning trends. The *performance weight* is then defined as $\boldsymbol{w}^{per} = \text{Softmax}\big([1/R_1^{tail}, 1/R_2^{tail}, \ldots, 1/R_N^{tail}]\big)$.

The final *sampling weight* $\boldsymbol{w}$ is formulated as $\boldsymbol{w} = \alpha \boldsymbol{w}^{sim} + (1-\alpha)\boldsymbol{w}^{per}$, where $\alpha$ is a hyperparameter to balance the two aspects. The CRA samples from each replay buffer $\mathcal{D}_i$ according to $\boldsymbol{w}$, ensuring a balanced and effective knowledge extraction and learning.

### 4.3 Overall Framework

The overall framework of CenRA is summarized in Algorithm 1. The CRA and policy agents are updated alternately and asynchronously, with the frequency of updating the CRA adjustable according to the actual situation. Sampling weights are calculated in real-time, using the most recently optimized encoders and the current learning performance, ensuring CRA continuously adjusts its focus to optimally balance knowledge extraction across multiple tasks.

The learned CRA acts as a robust knowledge pool, which is able to support new tasks by transferring knowledge through auxiliary reward signals. This is particularly beneficial in sparse-reward environments, as the knowledge rewards can guide the policy agents toward the correct direction and reduce exploration burden. Additionally, the CRA can be further optimized alongside new tasks in a continuous learning scheme that enhances adaptability and effectiveness in dynamic settings.

---

**Algorithm 1** Centralized Reward Agent based MTRL

---

**Require:** Multiple tasks $\{T_1, T_2, \ldots, T_N\}$.
**Require:** Policy agents $\{\mathcal{A}_1^{pol}, \mathcal{A}_2^{pol}, \ldots, \mathcal{A}_N^{pol}\}$.
**Require:** Centralized reward agent $\mathcal{A}^{rwd}$.
**Require:** Concatenated replay buffer $\mathcal{D} = \bigcup_{i=1}^{N} \mathcal{D}_i$.
 1: **for** each iteration **do**
 2:    **for** each task $T_i$ **do**
 3:       $(s_t, a_t, r_t^{env}, s_{t+1}, a_{t+1}) \sim \text{Interact}(\mathcal{A}_i^{pol}, T_i)$      ▷ Interact and collect one transition
 4:       $r_t^{knw} \sim \mathcal{A}^{rwd}(s_t, a_t)$        ▷ Sample an off-policy knowledge reward
 5:       $s_t^{rwd} = (s_t, a_t), s_{t+1}^{rwd} = (s_{t+1}, a_{t+1})$
 6:       $\mathcal{D}_i \leftarrow \mathcal{D}_i \cup \{(s_t^{rwd}, r_t^{knw}, r_t^{env}, s_{t+1}^{rwd}|T_i)\}$    ▷ Store the transition in corresponding $\mathcal{D}_i$
 7:       Update policy agent $\mathcal{A}_i^{pol}$       ▷ Update $\mathcal{A}_i^{pol}$ using backbone RL algorithm
 8:    **end for**
 9:    $\boldsymbol{w} = \alpha \boldsymbol{w}^{sim} + (1-\alpha)\boldsymbol{w}^{per}$       ▷ Calculate sampling weight
10:    $\{s_t^{rwd}, r_t^{knw}, r_t^{env}, s_{t+1}^{rwd}|T_i\}_{\mathcal{B}} \sim \mathcal{D}|\boldsymbol{w}$    ▷ Draw samples based on the sampling weight
11:    Update centralized reward agent $\mathcal{A}^{rwd}$
12: **end for**

---

## 5 Experiments

We conduct experiments in four MTRL domains as shown in Figure 2: the widely used **Meta-World** benchmark (including **ML10** with 10 tasks and **ML50** with 50 tasks) [Yu et al., 2020b], **2DMaze**, **3DPickup** [Chevalier-Boisvert et al., 2024], and **MujocoCar** [Ji et al., 2023]. All tasks, including those in Meta-World, are crafted to provide **sparse environmental rewards**, where the agent receives a reward of 1 only upon successful completion of the final objective, and 0 otherwise. The detailed task configurations are provided in Appendix A.

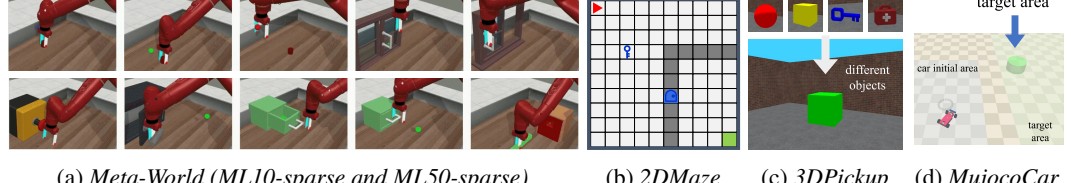

(a) *Meta-World (ML10-sparse and ML50-sparse)*   (b) *2DMaze*   (c) *3DPickup*   (d) *MujocoCar*

Figure 2: Environments with multiple tasks. (a) **Meta-World**: two sparse-reward versions are used: **ML10-sparse** and **ML50-sparse**, including diverse robotic manipulation tasks. (b) **2DMaze**: 2D maze tasks where the agent must pick up a key and then pass through a door to exit. (c) **3DPickup**: 3D maze tasks where the agent aims to navigate to and pick up different target objects at different locations. (d) **MujocoCar**: mujoco-based race car aims to navigate to different specified areas.

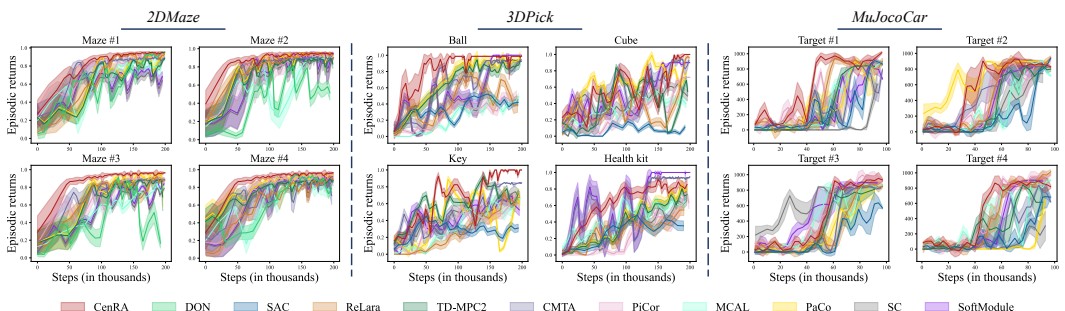

Figure 3: Comparison of CenRA with baselines in *2DMaze*, *3DPickup*, and *MujocoCar* domains.

## 5.1 Comparative Evaluation in MTRL

We benchmark CenRA against several state-of-the-art baselines: (a) the backbone RL algorithms of the policy agents: DQN [Mnih et al., 2015] for discrete control tasks and SAC [Haarnoja et al., 2018b] for continuous control tasks; (b) the ReLara algorithm [Ma et al., 2024a], which can be regarded as a decentralized variant of CenRA, where each policy agent is paired with a separate reward agent, without cross-task information sharing; (c) the TD-MPC2 algorithm [Hansen et al., 2024]; (d) the Contrastive Modules with Temporal Attention (CMTA) algorithm [Lan et al., 2023]; (e) the Policy Optimization and Policy Correction (PiCor) algorithm [Bai et al., 2023]; (f) the Multi-Critic Actor Learning (MCAL) algorithm [Mysore et al., 2022]; (g) the Parameter-compositional MTRL (PaCo) algorithm [Sun et al., 2022]; (h) the Shared-Critic (SC) algorithm [Zhang et al., 2021]; and (i) the MTRL with Soft Modularization (SoftModule) [Yang et al., 2020]. They are implemented by either the *CleanRL* library [Huang et al., 2022] or official codebases. Each task is trained with 10 different random seeds, and the average results are reported.

In the *Meta-World* domain, *ML10-sparse* provides 10 training tasks and 5 held-out test tasks, while *ML50-sparse* includes 45 training tasks and 5 test tasks. For the remaining domains, each consists of 4 training tasks and 1 test task. In this section, we evaluate the final returns achieved by the trained agents, averaged over all training tasks in each domain, as shown in Table 1. We additionally report the episodic returns and their standard errors throughout training in the *2DMaze*, *3DPickup*, and *MujocoCar* domains in Figure 3. To ensure a fair comparison, we adopt consistent hyperparameters (where applicable) and identical network architectures across all experiments; detailed configurations are provided in Appendix B.

We observe that CenRA consistently outperforms all baselines in three main aspects. First, it achieves the highest episodic returns in all tasks, demonstrating superior learning efficiency and faster convergence. Moreover, it demonstrates good stability and robustness, exhibiting fewer fluctuations and oscillations, especially after convergence, compared to other models. Notably, all tasks provide only sparse rewards, CenRA addresses this challenge through the auxiliary dense rewards with meaningful information, effectively guiding learning. This mechanism not only distinguishes CenRA from other structurally shared methods, but also provides a targeted solution to the sparse-reward problem. Second, while baselines like PiCor and MCAL often show uneven progress across different

Table 1: Episodic returns (mean ± standard error) of all trained agents tested over 100 episodes and averaged across all training tasks in each domain (↑ higher is better).

| Algorithm | ML10-sparse | ML50-sparse | 2DMaze | 3DPickup | MujocoCar |
|---|---|---|---|---|---|
| CenRA (ours) | **0.875 ± 0.121** | **0.755 ± 0.034** | **0.913 ± 0.023** | **0.880 ± 0.060** | **514.875 ± 0.675** |
| DQN/SAC | 0.256 ± 0.056 | 0.189 ± 0.012 | 0.645 ± 0.070 | 0.243 ± 0.048 | 198.000 ± 0.453 |
| ReLara | 0.674 ± 0.105 | 0.541 ± 0.057 | 0.803 ± 0.065 | 0.565 ± 0.088 | 429.800 ± 0.655 |
| TD-MPC2 | 0.823 ± 0.091 | 0.608 ± 0.032 | 0.884 ± 0.046 | 0.712 ± 0.051 | 505.341 ± 0.712 |
| CMTA | 0.787 ± 0.076 | 0.603 ± 0.026 | 0.753 ± 0.037 | 0.695 ± 0.043 | 480.187 ± 0.623 |
| PiCor | 0.865 ± 0.230 | 0.672 ± 0.123 | 0.818 ± 0.053 | 0.438 ± 0.085 | 437.550 ± 0.663 |
| MCAL | 0.842 ± 0.067 | 0.605 ± 0.055 | 0.885 ± 0.080 | 0.548 ± 0.068 | 369.200 ± 0.595 |
| PaCo | 0.854 ± 0.045 | 0.582 ± 0.022 | 0.834 ± 0.057 | 0.557 ± 0.072 | 421.210 ± 0.635 |
| SC | 0.556 ± 0.063 | 0.354 ± 0.023 | 0.798 ± 0.052 | 0.687 ± 0.038 | 400.254 ± 0.518 |
| SoftModule | 0.630 ± 0.042 | 0.423 ± 0.057 | 0.822 ± 0.076 | 0.486 ± 0.055 | 355.125 ± 0.594 |

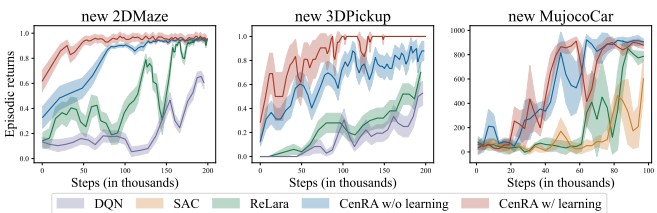
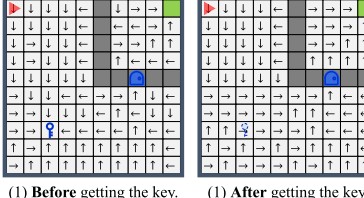

(a) Comparison of the learning performance of CenRA with the baselines in new tasks in the *2DMaze*, *3DPickup* and *MujocoCar* domains.

(b) Actions yielding the highest knowledge rewards in a new *2DMaze* task.

Figure 4: Experimental results for knowledge transfer to new tasks.

tasks within the same domain, CenRA maintains well-balanced performance by showing relatively consistent learning progress and minimal variability across each four-task groups. This ensures that no single task dominates or lags behind, which is crucial in multi-task learning. Third, the CRA effectively enhances knowledge sharing among tasks. This is evident from the comparison with ReLara, which uses independent reward agents and lacks the mechanism for knowledge exchange. By extracting and distributing insights from one task to another, the CRA improves the learning efficiency of individual tasks, highlighting the advantages of integrated knowledge management.

## 5.2 Knowledge Transfer to New Tasks

In this section, we assess the CRA's ability to transfer previously learned knowledge to unseen tasks. Specifically, we continue using the trained CRA model in Section 5.1, while initializing new policy agents to tackle new tasks from the same domain. These include 5 test tasks for *ML10-sparse* and *ML50-sparse*, and 1 test task for each of the remaining domains, none of which were encountered during the initial training. For the CenRA, we explore two scenarios: (1) the CRA continues to be optimized in collaboration with the new policy agent (CenRA w/ learning); and (2) only the policy agent is updated while the CRA remains fixed, relying only on its previously acquired knowledge (CenRA w/o learning). We compare the two settings against the backbone algorithms and ReLara. In ReLara, the reward agent is trained anew without pre-learned knowledge. The results are presented in Figure 4a and Table 2.

We observe that CenRA with further learning achieved rapid convergence, mainly due to the CRA's ability to retain previously acquired knowledge while continuing to adapt to new tasks through ongoing optimization. Remarkably, even without any additional training, CenRA still outperforms both ReLara, which requires training a new reward agent, and the backbone algorithms, which lack additional information. This advantage stems from the CRA's ability to encode and transfer environment-relevant knowledge, which can then be directly reused by new policy agents to guide their learning. Such knowledge transfer is particularly critical in our experiments involving challenging sparse-reward tasks. Without any external knowledge, learning would require extensive exploration. However, the CRA provides meaningful dense rewards that significantly accelerate the learning process, even during the initial phases.

Table 2: Episodic returns (mean $\pm$ standard error) of all trained agents in the new tasks, tested over 100 episodes in each domain ($\uparrow$ higher is better).

| Algorithm | ML10-sparse | ML50-sparse | 2DMaze | 3DPickup | MujocoCar |
|---|---|---|---|---|---|
| CenRA w/ learning | **0.902 $\pm$ 0.021** | **0.824 $\pm$ 0.012** | **0.952 $\pm$ 0.010** | **0.963 $\pm$ 0.002** | **532.080 $\pm$ 1.610** |
| CenRA w/o learning | 0.887 $\pm$ 0.011 | 0.809 $\pm$ 0.009 | 0.894 $\pm$ 0.032 | 0.678 $\pm$ 0.003 | 524.727 $\pm$ 0.588 |
| ReLara | 0.702 $\pm$ 0.086 | 0.612 $\pm$ 0.012 | 0.759 $\pm$ 0.056 | 0.263 $\pm$ 0.002 | 224.648 $\pm$ 0.492 |
| DQN/SAC | 0.228 $\pm$ 0.105 | 0.210 $\pm$ 0.034 | 0.263 $\pm$ 0.084 | 0.158 $\pm$ 0.003 | 129.055 $\pm$ 0.296 |

Table 3: Comparison of CenRA with ablation of different batch sampling control weights.

| Algo. | 2DMaze | | | | Var. $\downarrow$ ($\times 10^{-2}$) |
|---|---|---|---|---|---|
| | Maze #1 | Maze #2 | Maze #3 | Maze #4 | |
| **CenRA** ($\alpha = 0.5$) | **0.893 $\pm$ 0.033** | 0.908 $\pm$ 0.022 | **0.924 $\pm$ 0.020** | **0.932 $\pm$ 0.020** | 0.021 |
| CenRA ($\alpha = 0.25$) | 0.889 $\pm$ 0.031 | 0.901 $\pm$ 0.025 | 0.915 $\pm$ 0.023 | 0.925 $\pm$ 0.024 | 0.065 |
| CenRA ($\alpha = 0.75$) | 0.891 $\pm$ 0.030 | 0.905 $\pm$ 0.024 | 0.918 $\pm$ 0.021 | 0.928 $\pm$ 0.022 | 0.049 |
| w/o $\boldsymbol{w}^{sim}$ ($\alpha = 0$) | 0.884 $\pm$ 0.033 | **0.922 $\pm$ 0.021** | 0.873 $\pm$ 0.041 | 0.820 $\pm$ 0.039 | 0.172 |
| w/o $\boldsymbol{w}^{per}$ ($\alpha = 1$) | 0.758 $\pm$ 0.062 | 0.884 $\pm$ 0.030 | 0.824 $\pm$ 0.052 | 0.867 $\pm$ 0.020 | 0.284 |
| w/o both | 0.632 $\pm$ 0.053 | 0.833 $\pm$ 0.041 | 0.629 $\pm$ 0.08 | 0.802 $\pm$ 0.054 | 1.235 |

| Algo. | 3DPickup | | | | Var. $\downarrow$ ($\times 10^{-2}$) |
|---|---|---|---|---|---|
| | Ball | Cube | Key | Health kit | |
| **CenRA** ($\alpha = 0.5$) | **0.951 $\pm$ 0.020** | 0.683 $\pm$ 0.090 | **0.795 $\pm$ 0.062** | 0.688 $\pm$ 0.067 | 1.570 |
| CenRA ($\alpha = 0.25$) | 0.942 $\pm$ 0.023 | 0.671 $\pm$ 0.091 | 0.782 $\pm$ 0.066 | 0.715 $\pm$ 0.061 | 1.650 |
| CenRA ($\alpha = 0.75$) | 0.938 $\pm$ 0.025 | 0.665 $\pm$ 0.095 | 0.775 $\pm$ 0.069 | 0.702 $\pm$ 0.065 | 1.723 |
| w/o $\boldsymbol{w}^{sim}$ ($\alpha = 0$) | 0.822 $\pm$ 0.065 | **0.702 $\pm$ 0.093** | 0.704 $\pm$ 0.072 | **0.887 $\pm$ 0.038** | 0.892 |
| w/o $\boldsymbol{w}^{per}$ ($\alpha = 1$) | 0.779 $\pm$ 0.072 | 0.404 $\pm$ 0.093 | 0.631 $\pm$ 0.080 | 0.438 $\pm$ 0.102 | 3.051 |
| w/o both | 0.811 $\pm$ 0.073 | 0.457 $\pm$ 0.058 | 0.483 $\pm$ 0.079 | 0.370 $\pm$ 0.079 | 3.796 |

| Algo. | MujocoCar | | | | Var. $\downarrow$ ($\times 10^{3}$) |
|---|---|---|---|---|---|
| | Target #1 | Target #2 | Target #3 | Target #4 | |
| **CenRA** ($\alpha = 0.5$) | **588.221 $\pm$ 0.732** | **549.337 $\pm$ 0.640** | 447.743 $\pm$ 0.672 | **474.320 $\pm$ 0.657** | 4.244 |
| CenRA ($\alpha = 0.25$) | 575.153 $\pm$ 0.740 | 538.912 $\pm$ 0.655 | 439.850 $\pm$ 0.680 | 462.116 $\pm$ 0.665 | 4.871 |
| CenRA ($\alpha = 0.75$) | 580.431 $\pm$ 0.735 | 542.765 $\pm$ 0.648 | 441.033 $\pm$ 0.675 | 468.529 $\pm$ 0.660 | 4.533 |
| w/o $\boldsymbol{w}^{sim}$ ($\alpha = 0$) | 319.926 $\pm$ 0.590 | 486.767 $\pm$ 0.712 | 332.506 $\pm$ 0.695 | 260.921 $\pm$ 0.544 | 9.288 |
| w/o $\boldsymbol{w}^{per}$ ($\alpha = 1$) | 320.887 $\pm$ 0.891 | 355.325 $\pm$ 0.677 | 344.145 $\pm$ 0.872 | 308.215 $\pm$ 0.723 | 0.457 |
| w/o both | 57.532 $\pm$ 0.352 | 257.010 $\pm$ 0.677 | **677.285 $\pm$ 0.540** | 77.635 $\pm$ 0.255 | 82.720 |

To further demonstrate CenRA's transferability, we select the *2DMaze* environment to visualize the knowledge provided by CRA when facing an unseen task. As shown in Figure 4b, we plot the directions of actions that yield the maximum knowledge reward at each position, categorized into two scenarios: before and after obtaining the key. While some guidance in peripheral regions may appear slightly misaligned, most states receive reasonable rewards that align with human understanding. This demonstrates the effectiveness of knowledge transfer and with such dense rewards, the agent's adaptation to new tasks is able to be well supported. In addition, to better understand the knowledge learned by the CRA, we provide a case study in Appendix C, where we visualize the CRA-provided rewards. This further verifies that the CRA is capable of capturing domain-relevant, task-specific, and semantically meaningful signals across different tasks.

## 5.3 Effect of Sampling Weight

We conduct experiments to understand the effects of the information synchronization mechanism in the CenRA. Specifically, we compare the full CenRA model against five variants: (a) and (b) CenRA with different values of the balance factor $\alpha$, i.e., $\alpha = 0.25$ and $\alpha = 0.75$, to examine the impact of different weight combinations; (c) CenRA without the similarity weight $\boldsymbol{w}^{sim}$ (i.e., $\alpha = 0$); (d) CenRA without the performance weight $\boldsymbol{w}^{per}$ (i.e., $\alpha = 1$); and (e) CenRA without the entire sampling weight. To better illustrate the differences among tasks and highlight the role of sampling

weights in task coordination and synchronization, we select the four-task domains, i.e., *2DMaze*, *3DPickup*, and *MujocoCar*. The results are shown in Table 3.

The results indicate that the two weights, which control the allocation of samples drawn from each policy agent's experiences, mainly influence the overall learning performance. Specifically, the absence of sampling weight leads to unbalanced learning outcomes, which is observed by the increased variance in episodic returns across four tasks. In contrast, when both weights are incorporated, the learning process becomes notably more stable, indicating that the joint consideration of task similarity and learning progress is essential for coordinated optimization. While the full CenRA model does not always achieve the lowest variance, it consistently outperforms the other three ablation models regarding overall system performance.

Both weights play essential roles in information synchronization, with the performance weight $w^{\mathrm{per}}$ having a more significant impact. It allows the CRA to focus more on policy agents that are underperforming or progressing slowly, ensuring balanced system-wide learning. Moreover, different choices of the balancing factor $\alpha$ emphasize distinct aspects of synchronization: a larger $\alpha$ highlights task similarity and promotes uniformity across related tasks, whereas a smaller $\alpha$ prioritizes compensating lagging tasks by amplifying the effect of $w^{\mathrm{per}}$. This flexible weighting further enhances stability and adaptability, demonstrating that considering the overall learning performance of the multi-task system is a central objective that CenRA seeks to achieve.

## 6 Discussion and Conclusion

We propose a novel framework CenRA that integrates reward shaping into multi-task reinforcement learning. The framework shares domain knowledge across tasks to improve learning efficiency and effectively addresses the sparse-reward challenge. Specifically, the centralized reward agent (CRA) functions as a knowledge pool, responsible for distilling and distributing knowledge across tasks. Furthermore, the information synchronization mechanism mitigates imbalances in knowledge distribution, ensuring optimal system-wide performance. Experiments demonstrate that dense knowledge rewards generated by the CRA effectively guide policy learning, leading to faster convergence than baseline methods. CenRA also demonstrates superior and robust transferability to new tasks.

CenRA's main limitation is its requirement for consistent state and action dimensions across tasks. Future work could explore preprocessing techniques to adapt the framework to varying task structures, broadening its applicability. Additionally, the fixed trade-off between similarity weight and performance weight may not be ideal. A more flexible approach, such as adaptive weight regulation, could further enhance the framework. Moreover, the performance weight might favor underperforming tasks to achieve overall balance, but could limit the performance ceiling of high-performing tasks, indicating the need for a more effective trade-off mechanism.

## Acknowledgment

This work was supported by an Academic Research Grant MOE-T1 251RES2408 and a Research Scholarship from the Ministry of Education, Singapore.

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

# A Mutli-Task Experimental Configurations

We conduct experiments in four domains with multiple tasks: *Meta-World*, *2DMaze*, *3DPickup*, and *MujocoCar*. The detailed configurations of each task are illustrated in Figure 5. The *Meta-World* tasks illustration is adapted from [Yu et al., 2020b].

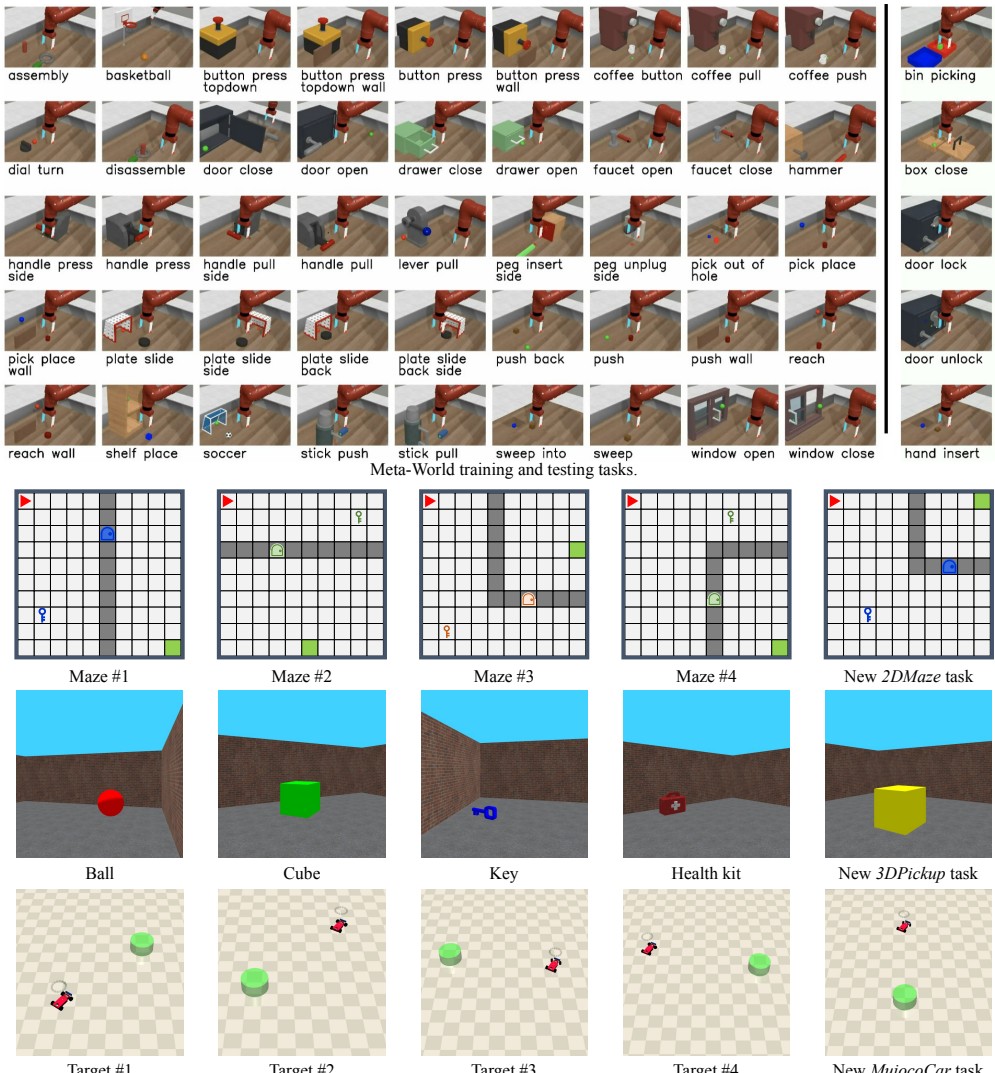

Figure 5: Illustration of multiple tasks in different domains in our experiments.

# B Network Structures and Hyperparameters

## B.1 Network Structures

Figure 6 illustrates the structures of all networks employed in our experiments.

## B.2 Hyperparameters

We have observed that CenRA demonstrated high robustness and was not sensitive to hyperparameter choices. Table 4 shows the hyperparameters we used in all the experiments.

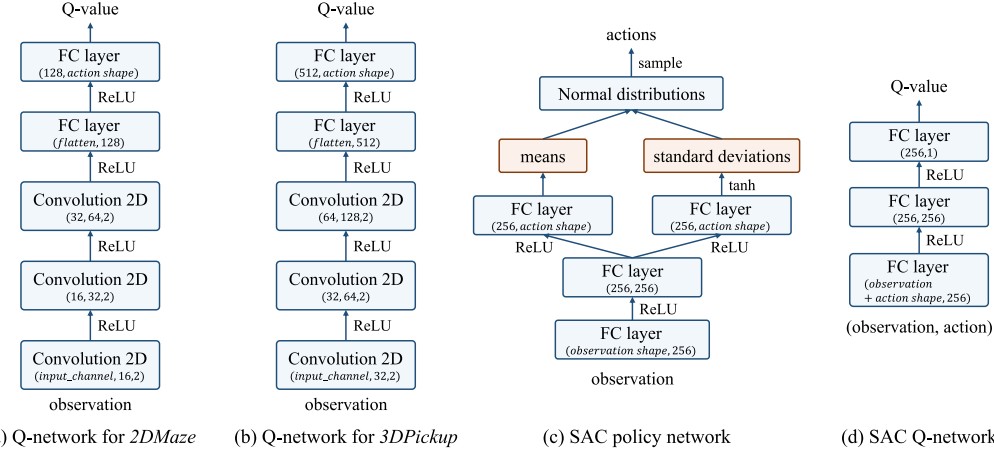

Figure 6: The structures of neural networks in our implementation.

Table 4: The hyperparameters of CenRA used in our experiments.

| Module | Hyperparameters | Values |
|---|---|---|
| Centralized Reward Agent $\mathcal{A}^{rwd}$ | discounted factor $\gamma$ | 0.99 |
| | batch size | 256 |
| | actor module learning rate | $3 \times 10^{-4}$ |
| | critic module learning rate | $1 \times 10^{-3}$ |
| | policy networks update frequency (steps) | 2 |
| | target networks update frequency (steps) | 1 |
| | target networks soft update weight $\tau$ | $5 \times 10^{-3}$ |
| | burn-in steps | 5000 |
| Policy Agent $\mathcal{A}_i^{pol}$ (DQN Agent) | knowledge reward weight $\lambda$ | 0.5 |
| | discounted factor $\gamma$ | 0.99 |
| | replay buffer size $|\mathcal{D}_i|$ | $1 \times 10^6$ |
| | batch size | 128 |
| | burn-in steps | 10000 |
| Policy Agent $\mathcal{A}_i^{pol}$ (SAC Agent) | knowledge reward weight $\lambda$ | 0.5 |
| | discounted factor $\gamma$ | 0.99 |
| | replay buffer size $|\mathcal{D}_i|$ | $1 \times 10^6$ |
| | batch size | 256 |
| | actor module learning rate | $3 \times 10^{-4}$ |
| | critic module learning rate | $1 \times 10^{-3}$ |
| | SAC entropy term factor $\alpha$ learning rate | $1 \times 10^{-4}$ |
| | policy networks update frequency (steps) | 2 |
| | target networks update frequency (steps) | 1 |
| | target networks soft update weight $\tau$ | $5 \times 10^{-3}$ |
| | burn-in steps | 10000 |

## B.3 Computing Resources

The experiments in this paper were conducted on a computing cluster, with the detailed hardware configurations listed in Table 5.

Table 5: The computing resources used in the experiments.

| Component | Specification |
|---|---|
| Operating System (OS) | Ubuntu 20.04 |
| Central Processing Unit (CPU) | 2x Intel Xeon Gold 6326 |
| Random Access Memory (RAM) | 256GB |
| Graphics Processing Unit (GPU) | 1x NVIDIA A100 20GB |
| Brand | Supermicro 2022 |

# C    What Has the Centralized Reward Agent Learned?

In this section, we visualize the learned *knowledge rewards* by the centralized reward agent $\mathcal{A}^{rwd}$ in the *2DMaze* environment. After training on the four tasks in Section 5.1 of the paper, we let the CRA generate the knowledge rewards for each action in every state and visualize the action direction that yields the maximum rewards, $a^* = \arg\max_a \pi^{rwd^*}(s_i, a), s_i \sim S$, in Figure 7.

The shaded areas in the figures represent regions within the real task that the agent cannot reach, as it cannot access the space behind the door without picking up the key. However, we forced the agent into these areas for evaluation. Outside the shaded regions, we observe that the CRA successfully learned meaningful knowledge rewards. Before picking up the key, the agent received the highest reward in the corresponding state when moving towards the key. Similarly, after picking up the key, the agent received the highest reward when moving towards the door and the final target. This demonstrates that in scenarios where the original environmental rewards are sparse, these detailed knowledge rewards can effectively guide the agent to converge more quickly.

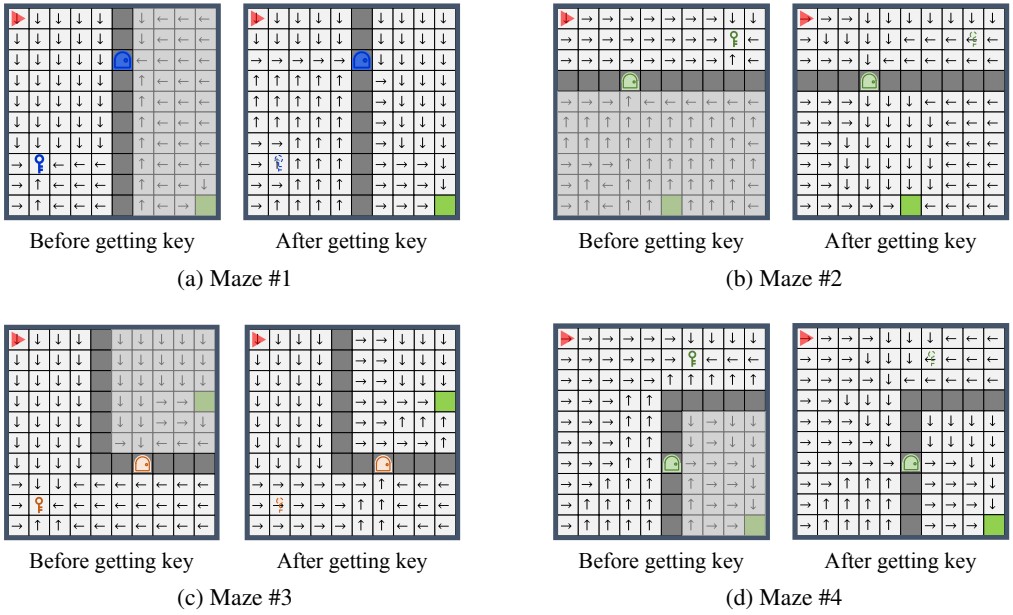

Figure 7: The actions yielding the maximum knowledge rewards in the four *2DMaze* tasks.

