# OpenReview forum: "Centralized Reward Agent for Knowledge Sharing and Transfer in Multi-Task Reinforcement Learning"
_NeurIPS.cc/2025/Conference — NeurIPS 2025 poster_

### Official Review · Reviewer_6NLb · 2025-06-10

**Clarity:** 3
**Significance:** 3
**Originality:** 3
**Rating:** 4
**Confidence:** 1

**Summary:**

This paper proposes a novel multi-task reinforcement learning (MTRL) framework called CenRA, which incorporates a Centralized Reward Agent (CRA) to distill and distribute task-specific knowledge across different policy agents through dense shaped rewards. The CRA learns from the experience of all policy agents and returns "knowledge rewards" that augment sparse environmental feedback. An information synchronization mechanism balances the CRA’s focus by considering task similarity and learning progress, ensuring fair and effective knowledge distribution. Experiments on discrete and continuous tasks, including Meta-World, 2DMaze, 3DPickup, and MujocoCar, demonstrate significant improvements in learning efficiency, task balance, and knowledge transfer compared to strong baselines.

**Questions:**

1. How does the CRA generalize when tasks differ slightly in state/action space dimensions or in reward scale? Could preprocessing help extend to those cases?

2. Is there any risk that the CRA introduces reward biases that harm long-term task-specific optimality?

3. How does CenRA perform in continual learning or lifelong RL settings, where tasks arrive sequentially?

4. Could the framework benefit from integrating meta-learning techniques to adapt the CRA or policy agents more efficiently?

5. Have you explored ways to automatically adjust the α parameter in the synchronization mechanism instead of fixing it manually?

6. What is the computational cost and training time increase introduced by CRA compared to decentralized baselines like ReLara?

Since this article is outside the scope of my research, I will refer to the opinions of other reviewers in the future.

**Ethical Concerns:**

["NO or VERY MINOR ethics concerns only"]

**Limitations:**

yes

**Quality:**

3

**Strengths And Weaknesses:**

Strengths

1. Clear Motivation and Problem Relevance: Tackles the widely acknowledged issue of sparse rewards in MTRL, providing a practical and novel reward-centric solution.

2. Innovative Framework Design: The separation between reward generation (CRA) and control (policy agents) is conceptually clean and modular, allowing easy integration into existing RL pipelines.

3. Knowledge Distillation via Dense Rewards: Using learned rewards as a medium for inter-task knowledge transfer is both elegant and effective.

4. Information Synchronization Mechanism: The combination of task similarity and learning performance for adaptive sampling ensures balanced system-wide learning.

5. Strong Empirical Results: Demonstrates consistent improvements across multiple environments and outperforms competitive baselines.

6. Knowledge Transfer to Unseen Tasks: Shows clear evidence that CRA retains and transfers useful knowledge, accelerating learning in new tasks.

7. Reproducibility: Detailed algorithm descriptions, code availability, hyperparameter listings, and experimental setups are thoroughly presented.

Weaknesses

1. Assumption of Homogeneous Input/Output Shapes: CRA requires the same state and action shape across tasks, limiting its applicability to more heterogeneous domains.

2. Fixed Trade-off Parameter ($\alpha$) for Sampling Weights: The performance of the synchronization mechanism relies on a manually tuned α, which may not generalize well across tasks or domains.

3. Computational Overhead: Maintaining and updating a centralized CRA while running multiple policy agents and computing cross-attention-based similarity adds nontrivial complexity.

4. Limited Theoretical Analysis: Although the empirical results are strong, there is no theoretical discussion on the optimality or convergence of the CRA-generated rewards.

5. CRA as a Bottleneck: In large-scale settings, the CRA may become a performance bottleneck due to its central role in reward generation and shared memory access.

6. Underexplored Comparisons: More ablations comparing CenRA to existing multi-task policy distillation or attention-based modular methods would strengthen the position of the proposed architecture.

---

> ### Author Rebuttal · Authors · 2025-07-31
>
> Dear reviewer,
>
> Thank you for your feedback. We respond to your comments as follows:
>
> ## Weakness 1 & Question 1
>
> Regarding the assumption of homogeneous input/output shapes, the current design requires all tasks to share the same state and action shapes. This limitation is already discussed in the conclusion.
>
> However, we also propose a potential solution to extend CenRA to heterogeneous multi-task settings by **padding the state and action vectors** of tasks with shorter dimensions, aligning them to a unified shape. This idea has been adopted in previous works such as *TD-MPC2* (Hansen, Nicklas et al. ICLR 2024) and *KTM-DRL* (Xu, Zhiyuan, et al. NeurIPS 2020). We believe that this approach can be effectively applied to CenRA, allowing it to handle tasks with varying state/action dimensions.
>
> ## Weakness 2 & Questions 5
>
> Regarding the trade-off parameter $\alpha$ in the sampling weights, we would like to clarify that $\alpha$ mainly aims to balance the contributions of the two weights. In our experiments, we adopt a fixed $\alpha=0.5$, which assigns equal importance to both components, and ensures the final sampling weight in the range $[0,1]$: $\omega=0.5 \omega^{sim} + 0.5 \omega^{per}$.
>
> We want to highlight that, the $\alpha$ is fixed, but the two weights $\omega^{sim}$ and $\omega^{per}$ are dynamically computed along with the training process, where $\omega^{sim}$ captures structural disparities across tasks, allowing CRA to attend more to outlier tasks, and $\omega^{per}$ monitors the real-time learning progress of each agent, allowing CRA to attend more to underperforming tasks. Since the two weights are dynamically computed, fixing $\alpha$ does not limit the flexibility and actually is a more reasonable choice. This is also a straightforward way for users to set the relative importance of the two criteria if needed.
>
> Finally, to provide a more comprehensive analysis, beyond the ablation results in Table 3, we have conducted an additional study analyzing the effect of different trade-off $\alpha$. The results indicate that while both weights significantly contribute to performance, the framework remains robust across a range of reasonable $\alpha$ values, demonstrating the flexibility of the proposed design.
>
> Table 1: Average performance per domain (**higher is better**)
>
> |Algo.|*ML10-sparse*|*ML50-sparse*|*2DMaze*|*3DPickup*|*MujocoCar*|
> |:---:|:---:|:---:|:---:|:---:|:---:|
> |CenRA($\alpha=0.5$)|0.88±0.12|0.76±0.03|0.91±0.02|0.88±0.06|514.88±0.68|
> |CenRA($\alpha=0.25$)|0.85±0.11|0.74±0.04|0.90±0.03|0.87±0.06|498.21±0.71|
> |CenRA($\alpha=0.75$)|0.87±0.12|0.75±0.04|0.91±0.03|0.86±0.06|507.45±0.69|
> |w/o $w^{sim}$|0.79±0.10|0.67±0.05|0.87±0.03|0.78±0.07|349.03±0.64|
> |w/o $w^{per}$|0.73±0.08|0.62±0.05|0.83±0.04|0.56±0.09|332.13±0.79|
> |w/o both|0.68±0.07|0.55±0.04|0.72±0.06|0.53±0.08|258.85±0.46|
>
> Table 2: The variances across multiple tasks per domain (**lower is better**)
>
> |Algo.|*ML10-sparse*|*ML50-sparse*|*2DMaze*|*3DPickup*|*MujocoCar*|
> |:---:|:---:|:---:|:---:|:---:|:---:|
> |CenRA ($\alpha=0.5$)|**0.08**|**0.16**|**0.02**|1.57|4.24|
> |**CenRA($\alpha=0.25$)**|0.18|0.22|0.03|1.82|6.88|
> |**CenRA($\alpha=0.75$)**|0.17|0.21|0.03|1.65|5.53|
> |w/o $w^{sim}$|0.25|0.18|0.17|**0.89**|9.29|
> |w/o $w^{per}$|0.35|0.24|0.28|3.05|**0.46**|
> |w/o both|0.40|0.68|1.24|3.80|82.7|
>
> ## Weakness 3 & Question 6
>
> Regarding the computational cost, the training time increase introduced by CRA, and the comparison with decentralized baselines like ReLara, we want to share the following highlights:
>
> 1. **Memory cost of agents**: Assume there are $N$ tasks and the reward agent (RA) and policy agent (PA) have roughly equal model sizes. In ReLara, each task maintains its independent RA, thus, the total memory cost is proportional to $2N$ (i.e., $N$ PAs + $N$ RAs). In contrast, CenRA uses only a single centralized RA shared across all tasks, resulting in a total memory cost of $N+1$, which is significantly more efficient as the number of tasks grows. Even compared to the single-task setting, where each task is paired with one RL agent (i.e., $N$ agents in total), CenRA achieves comparable efficiency by requiring only one additional RA.
> 2. **Replay buffer memory**: Both CenRA and ReLara train their RAs by sampling from the replay buffers of PAs, so the memory overhead related to experience storage is equivalent.
> 3. **Training time cost**: Assuming that training one agent (policy or reward) takes approximately the same amount of time, decentralized ReLara requires training $2N$ agents, while CenRA trains only $N + 1$ agents. Therefore, CenRA achieves a clear advantage in overall training time, especially in large-scale task settings.
> 4. **Resources distillation**: Fundamentally, CenRA is a centralized extension of ReLara, designed to distill and share task-relevant reward knowledge. In decentralized ReLara, each RA may redundantly learn overlapping knowledge across tasks. CenRA addresses this redundancy by unifying reward modeling into a single agent, thereby enabling more effective knowledge sharing while reducing both memory and computational burden.
>
> Overall, CenRA is more efficient than decentralized approaches like ReLara.
>
> ## Weakness 4 & Question 2
>
> Regarding the theoretical analysis of the CRA-generated rewards on optimality and convergence, as well as the risk of introducing reward biases to the original task-specific rewards, we would like to highlight: CenRA is an extension of ReLara, which has already demonstrated the ability to learn meaningful and task-aligned reward structures. ReLara ensures that the shaped rewards are consistent with the original task-specific objectives and do not interfere with the long-term optimality of the learned policy. Given that CenRA follows a similar principle, we believe the risk of introducing harmful reward bias remains minimal. Furthermore, all experiments in our paper evaluate the **original environmental rewards**, further validating that CenRA improves performance in terms of the original reward structure.
>
> ## Weakness 5
>
> Regarding the concern that CRA may become a bottleneck in large-scale settings:
>
> 1. **Benefit from large-scale MTRL**: As the number of tasks increases, the CRA gains a greater advantage in capturing task correlations and extracting shared knowledge. The global perspective allows it to better model cross-task commonalities and provide more informative shaped rewards. That said, we acknowledge that in extremely large-scale or highly heterogeneous task settings, the CRA’s capacity could become a limiting factor. Addressing this scalability challenge could involve several strategies:
> - **Modular reward agents**: Dividing the CRA into multiple specialized agents, each focusing on a subset of tasks or task clusters.
> - **Task clustering**: Grouping similar tasks and allowing the CRA to operate at a higher level of abstraction, reducing the complexity of the reward generation process.
> 2. **On shared memory and data storage overhead**: CRA samples experiences directly from the replay buffers maintained by individual policy agents, without requiring additional memory, no matter how many tasks. However, we recognize that as the number of tasks grows, the cumulative memory of all replay buffers may become significant. In such cases, distributed storage or memory-efficient experience compression strategies could be employed to mitigate the issue.
>
> ## Weakness 6
>
> Regarding more comparisons to existing policy distillation and attention-based modular methods, we have conducted further comparisons: (1) **TD-MPC2** (Hansen, Nicklas et al., ICLR 2024); (2) **CMTA** (Lan, Siming et al., NeurIPS 2023); (3) **Shared-Critic (SC)** (Zhang, Gengzhi et al., ACML 2021);  (4) **Distral** (Yee Whye Teh et al., NeurIPS 2017). In which *Distral* is a representative policy distillation method, while *CMTA* is a recent attention-based method. The results are shown below:
>
> |Algorithm|*ML10-sparse*|*ML50-sparse*|*2DMaze*|*3DPickup*|*MujocoCar*|
> |:---:|:---:|:---:|:---:|:---:|:---:|
> |CenRA (ours)|0.88±0.12|0.76±0.03|0.91±0.02|0.88±0.06|514.88±0.68|
> |TD-MPC2|0.82±0.09|0.61±0.03|0.88±0.05|0.71±0.05|505.34±0.71|
> |CMTA|0.79±0.08|0.60±0.03|0.75±0.04|0.70±0.04|480.19±0.62|
> |SC w/o sampling|0.56±0.06|0.35±0.02|0.80±0.05|0.69±0.04|400.25±0.52|
> |SC w/ sampling|0.60±0.06|0.41±0.03|0.73±0.05|0.53±0.05|430.18±0.57|
> |Distral|0.51±0.07|0.41±0.02|0.66±0.06|0.66±0.03|380.10±0.50|
>
> The additional results further validate the effectiveness of CenRA, which consistently achieves strong or superior performance across all domains. The new results will be included in the revised manuscript.
>
> ## Question 3
>
> Regarding CenRA's performance in continual learning or lifelong RL settings, we want to highlight that they can be viewed as a special case of our CenRA framework, where tasks arrive sequentially rather than concurrently. CenRA is well-suited to this setting, which is evidenced in our experiments (Section 5.2). The CRA is capable of transferring learned knowledge from previous tasks to unseen ones, and more importantly, CRA can be continuously updated with new tasks. This enables CenRA to effectively handle sequential task settings.
>
> ## Question 4
>
> Regarding the potential benefit of integrating meta-learning techniques, we appreciate the reviewer’s insightful suggestion. We agree that meta-learning holds strong potential for enhancing both the adaptability and efficiency of CenRA, e.g., meta-learning could be integrated by:
> * Learning a meta-reward function that adapts more quickly to new tasks with limited data
> * Employing meta-initialization strategies (e.g., MAML-style) to speed up policy agent learning on new tasks
> * Introducing a task encoder to learn latent task embeddings, which could inform both reward generation and sampling strategies.
>
> We believe integrating meta-learning is a promising future research direction.
>
> Once again, thank you for your comments. We hope our responses have addressed your concerns.

---

> ### Author Response · Authors · 2025-08-07
> **Follow-up on Rebuttal and Feedback**
>
> Dear Reviewer 6NLb,
>
> Thank you again for your initial review. We have carefully addressed all your concerns and questions in our rebuttal, and we hope our responses have effectively clarified the issues you raised.
>
> As the discussion phase is coming to a close, we would greatly appreciate your feedback, whether our responses have resolved your concerns, or if there are further points you'd like to discuss. If you find our replies satisfactory, we would be grateful if you could consider updating your score accordingly.
>
> Best regards,
>
> Authors of Submission #20587

---

> > ### Comment · Reviewer_6NLb · 2025-08-07
> >
> > Thanks to the authors for the responses. I will keep my positive scores.

---

> > > ### Author Response · Authors · 2025-08-07
> > > **Thank You for Maintaining Your Positive Evaluation**
> > >
> > > Dear Reviewer,
> > >
> > > We are very happy that our response addressed your concerns and that you will maintain your positive scores for our paper. We truly appreciate your support and constructive feedback.
> > >
> > > Best regards,
> > >
> > > Authors of Submission #20587

---

### Official Review · Reviewer_vuKu · 2025-06-26

**Clarity:** 3
**Significance:** 3
**Originality:** 2
**Rating:** 5
**Confidence:** 3

**Summary:**

The paper follows the idea of knowledge sharing by reward shaping and propose a framework to apply it to multitask RL problems. The proposed framework utilizes $N$ policies to solve $N$ tasks, and a centralized policy that shapes the rewards of these policies. To train the centralized policy, the authors use a weighted sampling approach, which is later justified in ablation experiments. The framework is evaluated in several benchmarks, demonstrating the performance of the proposed framework comparing to various baselines. The generalization ability of the centralized reward agent to unseen tasks is also demonstrated.

**Questions:**

1. Did you experiment in multi domain setting (i.e., tasks can be in different domains)?
2. Is the reward balanced (same scale) across tasks among all the presented simulations? Do you have a suggestion how to adapt for such cases?
3. for the DQN/SAC baselines, did you use a single instance for all tasks or one instance per task?

**Ethical Concerns:**

["NO or VERY MINOR ethics concerns only"]

**Final Justification:**

The authors addressed my concerns; added a baseline to demonstrate the contribution of the reward sampling method, included more MTRL baselines, and clarified the seeds issue. In light of the other reviews and responses, I raise my score to accept.

**Limitations:**

Limitations of this work are mostly discussed, I think that a few sentences regarding working with tasks that have different reward scale should be included.

**Paper Formatting Concerns:**

No formatting concerns

**Quality:**

3

**Strengths And Weaknesses:**

## Strengths
1. The framework is simple and effective, it allows using any backbone RL algorithm, where each task could be in different domain.
2. Extensive experiments across various domains.
3. The paper is well-written, Figure 1 and Algorithm 1 helps the reader to understand the method.
4. Justifies the proposed sampling method with an ablation experiment.

## Weaknesses
1. The ablation experiment validates the usage of the proposed weighted sampling, although it raises a concern regarding the contribution of the reward-generating agent. Consider adding the following baseline: shared critic that depends on the task info as well, trained with/without the proposed weighted sampling approach.
2. More MTRL/ contextual RL methods are missing, most of the baselines are not intended for this setting. I am not sure which methods are considered SOTA in this setting, but there are more recent works that fit for this setting.
3. As I understand, the reported results were generated from a single training of each baseline. It is important to evaluate over a few instances of each baseline, since the training process may vary for different random seeds. This is a major concern since the paper relies on the empirical evidence in these experiments.

---

> ### Author Rebuttal · Authors · 2025-07-31
>
> Dear reviewer,
>
> Thanks for your comments, and below we provide detailed responses.
>
> ## Weaknesses
>
> **Weakness 1**:
>
> > The ablation experiment validates the usage of the proposed weighted sampling, although it raises a concern regarding the contribution of the reward-generating agent. Consider adding the following baseline: shared critic that depends on the task info as well, trained with/without the proposed weighted sampling approach.
>
> Regarding the contribution of the reward agent and suggested shared-critic baseline:
>
> We have included the *Shared-Critic* framework for MTRL as an additional baseline [3]. The author-proposed method doesn't have weighted sampling, so we compare with both the original shared-critic method (SC w/o sampling) and a modified version that incorporates our proposed weighted sampling approach (SC w/ sampling). The results are shown as follows:
>
> | Algorithm | *ML10-sparse* | *ML50-sparse* | *2DMaze* | *3DPickup* | *MujocoCar* |
> | :---: | :---: | :---: | :---: | :---: | :---: |
> | CenRA (ours)| **0.875 ± 0.121** | **0.755 ± 0.034** | **0.913 ± 0.023** | **0.880 ± 0.060** | **514.875 ± 0.675** |
> | **Shared-Critic (SC w/o sampling)** | 0.556 ± 0.063 | 0.354 ± 0.023 | 0.798 ± 0.052 | 0.687 ± 0.038 | 400.254 ± 0.518 |
> | **Shared-Critic (SC w/ sampling)** | 0.604 ± 0.058 | 0.407 ± 0.031 | 0.725 ± 0.049 | 0.526 ± 0.045 | 430.179 ± 0.567 |
>
> From the results, our CenRA method consistently outperforms SC. We highlight two key advantages:
>
> 1. **Reward-Based Knowledge Transfer**: CenRA used a centralized reward agent to share knowledge by generating *dense and informative rewards* for each policy agent. This allows CenRA to provide *step-wise guidance*, which is particularly beneficial in challenging sparse-reward environments. In contrast, SC's shared critic is still trained only on sparse rewards, so it requires extensive exploration for the actors to learn effectively.
> 2. **Contribution of the Reward Agent**: Even with our proposed weighted sampling, the performance of SC is still lower than CenRA, indicating that the centralized reward agent plays a crucial role in enhancing knowledge transfer. The weighted sampling helps balance the contributions of different tasks, but it does not fully compensate for the lack of a dedicated reward shaping mechanism.
>
> The new experiments will be added into the revised manuscript.
>
> [3] Zhang, Gengzhi et al., "Multi-task Actor-Critic with Knowledge Transfer via a Shared Critic". In Proceedings of the 13th Asian Conference on Machine Learning (Vol. 157, pp. 580–593). PMLR.
>
> **Weakness 2**:
>
> > More MTRL/ contextual RL methods are missing, most of the baselines are not intended for this setting. I am not sure which methods are considered SOTA in this setting, but there are more recent works that fit for this setting.
>
> Regarding the concern that more recent MTRL methods are missing, we would like to emphasize that our paper already includes several representative and competitive baselines. Specifically, we compared against PiCor (AAAI 2023), MCAL (ICLR 2022), PaCo (NeurIPS 2022), and SoftModule (NeurIPS 2022), all of which are recognized as strong MTRL algorithms.
>
> In addition, we have conducted further comparisons with several more recent and relevant baselines: (1) **TD-MPC2** (ICLR 2024) [1]; (2) **CMTA** (NeurIPS 2023) [2]; (3) **Shared-Critic (SC)** (ACML 2021) [3]; (4) **Distral** (NeurIPS 2017) [4]. In which the TD-MPC2 and CMTA can be considered as SOTA methods in MTRL, while the SC and Distral are also two representative baselines for multi-task RL. The results of these additional baselines are summarized in the following table:
>
> | Algorithm | *ML10-sparse* | *ML50-sparse* | *2DMaze* | *3DPickup* | *MujocoCar* |
> | :---: | :---: | :---: | :---: | :---: | :---: |
> | CenRA (ours)| **0.875 ± 0.121** | **0.755 ± 0.034** | **0.913 ± 0.023** | **0.880 ± 0.060** | **514.875 ± 0.675** |
> | DQN/SAC | 0.256 ± 0.056 | 0.189 ± 0.012 | 0.645 ± 0.070 | 0.243 ± 0.048 | 198.000 ± 0.453 |
> | ReLara | 0.674 ± 0.105 | 0.541 ± 0.057 | 0.803 ± 0.065 | 0.565 ± 0.088 | 429.800 ± 0.655 |
> | PiCor | 0.865 ± 0.230 | 0.672 ± 0.123 | 0.818 ± 0.053 | 0.438 ± 0.085 | 437.550 ± 0.663 |
> | MCAL | 0.842 ± 0.067 | 0.605 ± 0.055 | 0.885 ± 0.080 | 0.548 ± 0.068 | 369.200 ± 0.595 |
> | PaCo | 0.854 ± 0.045 | 0.582 ± 0.022 | 0.834 ± 0.057 | 0.557 ± 0.072 | 421.210 ± 0.635 |
> | SoftModule | 0.630 ± 0.042 | 0.423 ± 0.057 | 0.822 ± 0.076 | 0.486 ± 0.055 | 355.125 ± 0.594 |
> | TD-MPC2 | 0.823 ± 0.091 | 0.608 ± 0.032 | 0.884 ± 0.046 | 0.712 ± 0.051 | 505.341 ± 0.712 |
> | CMTA | 0.787 ± 0.076 | 0.603 ± 0.026 | 0.753 ± 0.037 | 0.695 ± 0.043 | 480.187 ± 0.623 |
> | Shared-Critic (SC w/o sampling) | 0.556 ± 0.063 | 0.354 ± 0.023 | 0.798 ± 0.052 | 0.687 ± 0.038 | 400.254 ± 0.518 |
> | Shared-Critic (SC w/ sampling) | 0.604 ± 0.058 | 0.407 ± 0.031 | 0.725 ± 0.049 | 0.526 ± 0.045 | 430.179 ± 0.567 |
> | Distral | 0.509 ± 0.072 | 0.406 ± 0.020 | 0.663 ± 0.062 | 0.655 ± 0.033 | 380.098 ± 0.499 |
>
> The experimental results from these newly added baselines further validate the effectiveness of CenRA, which consistently achieves strong or superior performance across all evaluated domains. These new results will be included in the revised version of the manuscript.
>
> [1] Hansen, Nicklas et al. "TD-MPC2: Scalable, Robust World Models for Continuous Control." The 12th International Conference on Learning Representations (2024).
>
> [2] Lan, Siming et al., "Contrastive Modules with Temporal Attention for Multi-Task Reinforcement Learning". In Proceedings of the 37th International Conference on Machine Learning (2023). PMLR.
>
> [3] Zhang, Gengzhi et al., "Multi-task Actor-Critic with Knowledge Transfer via a Shared Critic". In Proceedings of the 13th Asian Conference on Machine Learning (Vol. 157, pp. 580–593). PMLR.
>
> [4] Yee Whye Teh et al., "Distral: Robust multitask reinforcement learning". Advances in Neural Information Processing Systems 30 (2017): 4496-4506.
>
> **Weakness 3**:
>
> > As I understand, the reported results were generated from a single training of each baseline. It is important to evaluate over a few instances of each baseline, since the training process may vary for different random seeds. This is a major concern since the paper relies on the empirical evidence in these experiments.
>
> Regarding the evaluation with different random seeds, in our experiments, we indeed run multiple instances with different random seeds for each setting and baseline. This has been stated in our paper (**Line 247**): *"Each task is trained with 10 different random seeds, and the average results are reported."*.
>
>
> ## Questions
>
> > 1. Did you experiment in multi-domain setting (i.e., tasks can be in different domains)?
>
> Regarding the multi-domain setting: our experiments are conducted under the standard MTRL setting, where tasks are generally assumed to share the same domain but differ in their specific configurations, targets, and other factors. For example, in the Meta-World benchmark, all tasks involve robotic manipulation but vary in terms of the objects to be grasped or the types of interactions required. These tasks differ in observations and targets but still reside within a consistent domain (e.g., robotic control).
>
> In contrast, *multi-domain* settings where tasks span across fundamentally different domains are not usually considered in MTRL, (e.g., learning robotic manipulation and autonomous driving simultaneously). Such tasks involve distinct fundamental knowledge, making knowledge sharing less meaningful or even detrimental. Designing a unified framework for multi-domain MTRL remains an open and challenging problem, which we consider beyond the scope of this work.
>
> > 2. Is the reward balanced (same scale) across tasks among all the presented simulations? Do you have a suggestion how to adapt for such cases?
>
> Regarding the reward scale across tasks, we would like to clarify that currently, we are using the standard MTRL setting, as the tasks are defined within the same domain, their reward functions are consistent in numerical scale, e.g, there are **no cases** where some tasks yield rewards in the range $[0, 1]$ while others operate in $[0, 100]$. However, considering that in more general settings where tasks may have different reward magnitudes, we suggest applying a normalization wrapper to rescale the rewards of all tasks to a common range before computing the performance weights, which can be easily implemented by using `gym.wrappers.TransformReward` for standard OpenAI Gym environments.
>
> > 3. For the DQN/SAC baselines, did you use a single instance for all tasks or one instance per task?
>
> Regarding the DQN/SAC baselines, we assume that they are single-task settings, so we use **one independent instance of the DQN or SAC agent per task**. Each task is trained separately by assuming that they don't share any knowledge across tasks.
>
> Thanks again, and we hope our response has addressed your concerns.

---

> > ### Comment · Reviewer_vuKu · 2025-08-06
> >
> > I thank the authors for the comprehensive response, and the experiments they have conducted.
> > Most of my concerns were addressed, and with the changes that the authors suggested, I tend to raise my score accordingly. I am still waiting for the other reviewers, to make a better assessment, as they have raised more issues.

---

> > > ### Author Response · Authors · 2025-08-06
> > > **Thank You and Looking Forward to Your Raised Scores**
> > >
> > > Dear reviewer,
> > >
> > > We are glad to hear that our response has fully addressed your concerns. We also note that reviewer **UzuT** mentioned that his/her concerns have been resolved and that he/she is ready to provide a more positive evaluation of our paper. **We look forward to your update on the score.**
> > >
> > > We sincerely appreciate your time and support.
> > >
> > > Best regards, Authors of Submission #20587

---

> > > ### Author Response · Authors · 2025-08-07
> > > **Follow-Up on Score Update**
> > >
> > > Dear Reviewer,
> > >
> > > Thank you once again for your positive feedback on our responses.
> > >
> > > We would like to share that all other reviewers have responded favorably to our rebuttal and have updated their scores accordingly.
> > >
> > > As the reviewer-author discussion phase is drawing to a close, we sincerely hope you will consider their feedback alongside our previous response and kindly proceed with updating your score.
> > >
> > > Thank you for your time and thoughtful engagement.
> > >
> > > Best regards,
> > >
> > > Authors of Submission #20587

---

### Official Review · Reviewer_UzuT · 2025-07-02

**Clarity:** 4
**Significance:** 2
**Originality:** 3
**Rating:** 4
**Confidence:** 4

**Summary:**

This paper explores the use of reward shaping for multi-task reinforcement learning via a centralised reward agent. In overly simple words, CenRA is an extension of the reward shaping methodology to multi-task reinforcement learning. However, behind this "simple" sentence, there is a novel idea that is worth exploring.

Indeed, while reward shaping via an assistant reward agent is not a new concept, the co-articulation of this paradigm with multi-task reinforcement learning has not been explored before to the best of my knowledge, and could be done in very different ways. The authors propose to have a shared reply buffer (to reduce memory usage and simplify the code) as well as adaptive strategies to weight the various tasks depending on their similarity and the learning performance of the associated task-specific agents.

Experiments are conducted and reported in several environment sets, exhibiting consistent results and validating the intuition behind the paper.

**Questions:**

The core idea of the paper is to adopt the reward assistant agent paradigm for multi-task reinforcement learning. In this regard, the main methodological contribution should be in this "adoption". The authors have chosen fairly simple weight models and weight combination strategies (which I see as a positive thing).

However, there is not much discussion on other modelling possibilities. Weights and their combinations can be modelled in very different ways, and this can have a severe impact on the experiments. I wonder if the authors carried out an in-depth exploration.

In addition, regarding the task similarity weights, the task centroid "c" could be zero (or arbitrarily close to zero), thus opening the door to strong numerical stabilities. I would like the authors to comment on that point.

Likewise, for the task performance weights, the numerical range of the rewards of the various tasks is potentially very different (there are no constraints whatsoever announced by the authors), and therefore using the average past rewards to re-weight the replay buffer can be very problematic since tasks that naturally have a large reward would be visited less often, but for the wrong reason.

Overall, I believe that the main contribution of the paper (these weights and weight combination strategies) deserve more exploration both in terms of internal comparison performance as well as in terms of numerical instabilities.

**Ethical Concerns:**

["NO or VERY MINOR ethics concerns only"]

**Final Justification:**

I updated my rating from Borderline Reject to Borderline Accept accordingly to the authors rebutal.

**Limitations:**

Yes

**Paper Formatting Concerns:**

Nothing to report.

**Quality:**

3

**Strengths And Weaknesses:**

Strengths:
- The idea has not been explored before.
- The model design is simple and takes into account memory/code factors.
- The paper is well written and easy to understand.

Weaknesses:
- The motivation for which only similarity and learning weights are chosen is unclear, as it is the way to combine those weights.
- The intuition behind how to compute the weights is clear, but this computation is potentially prone to numerical instabilities.
These weaknesses are an issue because the weighting strategy is the main methodological contribution of the paper.

---

> ### Author Rebuttal · Authors · 2025-07-31
>
> Dear reviewer,
>
> Thank you for your comments. We have addressed your concerns in detail, along with the identified weaknesses and questions.
>
> ## Weakness 1 and Question 1
>
> > (Weakness) The motivation for which only similarity and learning weights are chosen is unclear, as it is the way to combine those weights.
>
> > (Question) However, there is not much discussion on other modelling possibilities. Weights and their combinations can be modelled in very different ways, and this can have a severe impact on the experiments. I wonder if the authors carried out an in-depth exploration.
>
> Regarding the motivation for using similarity and performance weights in the information synchronization mechanism:
>
> 1. **Motivation**: Our design is motivated by a systematic consideration of two complementary aspects in MTRL. The **similarity weight** captures structural disparities across tasks, allowing the CRA to attend more to outlier tasks and promote effective knowledge sharing. The **performance weight** monitors the real-time learning progress of each agent, encouraging the CRA to allocate more attention to underperforming tasks. The combination of the two enables the system to simultaneously maintain *representation diversity* and *training balance*, achieving both broad knowledge distillation and efficient convergence.
>
> 2. **Justification**: These weights are not only intuitive and interpretable, but also computationally efficient, as they rely on existing quantities—latent features and recent reward statistics. Moreover, their individual importance is empirically validated in Section 5.3, where removing either component leads to performance drops and increased variance across tasks (see Table 3).
>
> 3. **Exploration of More Formulations**: We intentionally began with these two **representative and controllable factors** to establish a principled and effective baseline. We agree that other formulations are worth exploring. There are two potential directions for future work:
>
> * Learnable or adaptive weighting schemes;
> * Alternative performance proxies, such as TD-error, policy entropy, or gradient norms.
>
> 4. **More experimental results on different weights combinations**: Finally, beyond the ablation results in Table 3, we have conducted an additional study (see the followng table) analyzing the effect of different trade-off parameter $\alpha$. The results indicate that while both weights significantly contribute to performance, the framework remains robust across a range of reasonable $\alpha$ values, demonstrating the flexibility of the proposed design.
>
> Table 1: The average performance per domain of CenRA with different $\alpha$ sampling weights (**higher is better**).
>
> | Algo. | *ML10-sparse* | *ML50-sparse* | *2DMaze* | *3DPickup* | *MujocoCar* |
> | :---: | :---: | :---: | :---: | :---: | :---: |
> | CenRA ($\alpha = 0.5$) | **0.875 ± 0.121** | **0.755 ± 0.034** | **0.913 ± 0.023** | **0.880 ± 0.060** | **514.875 ± 0.675** |
> | **CenRA ($\alpha = 0.25$)** | 0.852 ± 0.115 | 0.738 ± 0.038 | 0.895 ± 0.028 | 0.865 ± 0.062 | 498.210 ± 0.705 |
> | **CenRA ($\alpha = 0.75$)** | 0.868 ± 0.125 | 0.749 ± 0.036 | 0.908 ± 0.025 | 0.862 ± 0.061 | 507.450 ± 0.690 |
> | w/o $w^{sim}$ ($\alpha = 0$) | 0.785 ± 0.095 | 0.670 ± 0.050 | 0.872 ± 0.033 | 0.778 ± 0.068 | 349.025 ± 0.635 |
> | w/o $w^{per}$ ($\alpha = 1$) | 0.730 ± 0.080 | 0.615 ± 0.045 | 0.832 ± 0.040 | 0.563 ± 0.085 | 332.125 ± 0.790 |
> | w/o both | 0.680 ± 0.070 | 0.550 ± 0.040 | 0.720 ± 0.055 | 0.530 ± 0.078 | 258.850 ± 0.458 |
>
>
> Table 2: The variances across multiple tasks per domain of CenRA with different $\alpha$ sampling weights (**lower is better**).
>
> | Algo. | *ML10-sparse* | *ML50-sparse* | *2DMaze* | *3DPickup* | *MujocoCar* |
> | :---: | :---: | :---: | :---: | :---: | :---: |
> | CenRA ($\alpha = 0.5$) | **0.08** | **0.16** | **0.02** | 1.57 | 4.24 |
> | **CenRA ($\alpha = 0.25$)** | 0.18 | 0.22 | 0.03 | 1.82 | 6.88 |
> | **CenRA ($\alpha = 0.75$)** | 0.17 | 0.21 | 0.03 | 1.65 | 5.53 |
> | w/o $w^{sim}$ ($\alpha = 0$) | 0.25 | 0.18 | 0.17 | **0.89** | 9.29 |
> | w/o $w^{per}$ ($\alpha = 1$) | 0.35 | 0.24 | 0.28 | 3.05 | **0.46** |
> | w/o both | 0.40 | 0.68 | 1.24 | 3.80 | 82.7 |
>
>
> ## Weakness 2 and Question 2
>
> > (Weakness) The intuition behind how to compute the weights is clear, but this computation is potentially prone to numerical instabilities. These weaknesses are an issue because the weighting strategy is the main methodological contribution of the paper.
>
> > (Question) In addition, regarding the task similarity weights, the task centroid "c" could be zero (or arbitrarily close to zero), thus opening the door to strong numerical stabilities. I would like the authors to comment on that point.
>
> Regarding the concern that the task centroid $\mathbf{c}$ could become zero (or arbitrarily close to zero), potentially causing numerical instabilities in the similarity computation, we would like to clarify that this issue has been explicitly considered in our design. To mitigate the risk of feature vectors pointing in opposing directions and canceling each other out, in our approach, all latent features $\mathbf{H}\_i$ are extracted from the activation outputs of neural network layers, specifically, ReLU activations in our implementation, ensuring that all elements of $\mathbf{H}\_i$ are non-negative. This effectively prevents cancellation effects and ensures that the resulting centroid $\mathbf{c}$ remains well-defined and numerically stable. The relevant implementation details can be found in the `get_features` function in `CenRA/Networks.py` within our submitted codes. We will further clarify this point in the revised manuscript.
>
> ## Question 3
>
> > Likewise, for the task performance weights, the numerical range of the rewards of the various tasks is potentially very different (there are no constraints whatsoever announced by the authors), and therefore using the average past rewards to re-weight the replay buffer can be very problematic since tasks that naturally have a large reward would be visited less often, but for the wrong reason.
>
> Regarding the concern on reward scale differences affecting the performance weights, we would like to clarify that in our current setup, as the multi-task environments are defined within the same domain, thus their reward functions are consistent in numerical scale, e.g, there are **no cases** where some tasks yield rewards in the range $[0, 1]$ while others operate in $[0, 100]$. Therefore, the recent average rewards used in the performance weighting are directly comparable across tasks. However, considering that in more general settings where tasks may have different reward magnitudes, we suggest applying a normalization wrapper to rescale the rewards of all tasks to a common range before computing the performance weights, which can be easily implemented by using `gym.wrappers.TransformReward` for standard OpenAI Gym environments.
>
> Thanks again and hope our response has addressed your concerns.

---

### Official Review · Reviewer_V8Sn · 2025-07-03

**Clarity:** 3
**Significance:** 2
**Originality:** 2
**Rating:** 5
**Confidence:** 3

**Summary:**

The paper proposes a Centralized Reward Agent based Multi-Task RL framework (CenRA), where a centralized reward agent learns the policy to predict knowledge rewards that optimize the accumulated environment rewards it will receive, and policy agents for each task update their own policies by any RL algorithm to optimize the augmented reward (addition of the environment reward and the scaled knowledge reward). CenRA shows empirically improved performance compared to prior work on Meta-World, 2DMaze, 3DPickup, and MujocoCar.

**Questions:**

As described in Weaknesses.

**Ethical Concerns:**

["NO or VERY MINOR ethics concerns only"]

**Final Justification:**

As the authors have addressed all of my concerns and most of the other reviewers' questions with evidence, I have raised my score accordingly.

**Limitations:**

yes

**Quality:**

3

**Strengths And Weaknesses:**

### Strengths
* The paper is well structured, easy to follow, and clear.
* The proposed method, including the core knowledge distillation and distribution, as well as the information synchronization mechanism (similarity weight and performance weight), is reasonable to mitigate potential issues in such a framework.
* The experiments are conducted on Meta-World benchmark and additional 3 environments, showing consistent improvements.
* Although I have some questions about completeness (as described in Weaknesses), the paper has included important analyses like an ablation study of the similarity weight and performance weight, knowledge transfer to unseen tasks, and the visualization of learned knowledge rewards.

### Weaknesses
My questions are mainly about the experiment design and analysis to validate some components’ effectiveness:
* Since the 2DMaze involves the least diverse tasks (only the position and colors are different, but the goals are the same; the unseen task even shares the same colors as one of the seen task) among the four benchmarks, the claim of the success of knowledge transfer in * Section 5.2 will be strengthened if the paper can show Meta-World’s visualization results.
* In addition to Table3 for sampling weight ablation study, can the authors also show the Meta-World’s results? If the number of tasks is a concern to list the table, it will also be beneficial to see the results averaged across tasks.

---

> ### Author Rebuttal · Authors · 2025-07-31
>
> Dear reviewer,
>
> Thanks for the comments and below we provide detailed responses.
>
> ## Weakness 1
>
> > Since the 2DMaze involves the least diverse tasks (only the position and colors are different, but the goals are the same; the unseen task even shares the same colors as one of the seen task) among the four benchmarks, the claim of the success of knowledge transfer in Section 5.2 will be strengthened if the paper can show Meta-World’s visualization results.
>
> Regarding the visualization of the Meta-World tasks in Section 5.2, due to response format constraints, we are unable to upload PDF files or share anonymized image links. Therefore, we present a tabular format to “visualize” the CRA-generated rewards for the *door-unlock* task in Meta-World, which involves multi-step manipulation and spatio-temporal reasoning. The table illustrates how the CRA assigns different rewards at various stages of task completion, such as approaching the door handle, aligning with it, and rotating it to unlock, etc., highlighting its ability to produce meaningful rewards. The real visualizations with the corresponding figures will be included in the revised manuscript.
>
> | Description of Task Stage | CRA Reward |
> | :---: | :---: |
> | Initialization | 0.001 |
> | Agent moves towards the door handle. | 0.122 |
> | Agent is close to the door handle. | 0.213 |
> | Agent's gripper is roughly aligned with the handle. | 0.458 |
> | Agent's gripper has successfully grasped the handle. | 0.463 |
> | Agent starts rotating the handle. | 0.702 |
> | Handle is partially rotated. | 0.785 |
> | Handle is fully rotated to the unlocked position. | 0.833 |
> | Agent pushes the door slightly open. | 0.857 |
> | Door is fully open. | 0.925 |
>
> ## Weakness 2
>
> > In addition to Table3 for sampling weight ablation study, can the authors also show the Meta-World’s results? If the number of tasks is a concern to list the table, it will also be beneficial to see the results averaged across tasks.
>
> Regarding the ablation study of sampling weights, due to the large number of tasks in the Meta-World benchmark (45 training tasks), we didn't present the task-wise results in the main paper. Here, we report the averaged results across all tasks within each domain to summarize the impact of different ablation settings. Additionally, we extend the analysis by introducing two more ablation settings with different trade-offs between the similarity weight and the performance weight by setting $\alpha = 0.25$ and $\alpha = 0.75$. The detailed results are presented in the table below. These results further confirm the effectiveness and robustness of our weighting strategy.
>
> Table 1: The average performance per domain of CenRA with different $\alpha$ sampling weights (**higher is better**).
>
> | Algo. | *ML10-sparse* | *ML50-sparse* | *2DMaze* | *3DPickup* | *MujocoCar* |
> | :---: | :---: | :---: | :---: | :---: | :---: |
> | CenRA ($\alpha = 0.5$) | **0.875 ± 0.121** | **0.755 ± 0.034** | **0.913 ± 0.023** | **0.880 ± 0.060** | **514.875 ± 0.675** |
> | **CenRA ($\alpha = 0.25$)** | 0.852 ± 0.115 | 0.738 ± 0.038 | 0.895 ± 0.028 | 0.865 ± 0.062 | 498.210 ± 0.705 |
> | **CenRA ($\alpha = 0.75$)** | 0.868 ± 0.125 | 0.749 ± 0.036 | 0.908 ± 0.025 | 0.862 ± 0.061 | 507.450 ± 0.690 |
> | w/o $w^{sim}$ ($\alpha = 0$) | 0.785 ± 0.095 | 0.670 ± 0.050 | 0.872 ± 0.033 | 0.778 ± 0.068 | 349.025 ± 0.635 |
> | w/o $w^{per}$ ($\alpha = 1$) | 0.730 ± 0.080 | 0.615 ± 0.045 | 0.832 ± 0.040 | 0.563 ± 0.085 | 332.125 ± 0.790 |
> | w/o both | 0.680 ± 0.070 | 0.550 ± 0.040 | 0.720 ± 0.055 | 0.530 ± 0.078 | 258.850 ± 0.458 |
>
> Table 2: The variances across multiple tasks per domain of CenRA with different $\alpha$ sampling weights (**lower is better**).
>
> | Algo. | *ML10-sparse* | *ML50-sparse* | *2DMaze* | *3DPickup* | *MujocoCar* |
> | :---: | :---: | :---: | :---: | :---: | :---: |
> | CenRA ($\alpha = 0.5$) | **0.08** | **0.16** | **0.02** | 1.57 | 4.24 |
> | **CenRA ($\alpha = 0.25$)** | 0.18 | 0.22 | 0.03 | 1.82 | 6.88 |
> | **CenRA ($\alpha = 0.75$)** | 0.17 | 0.21 | 0.03 | 1.65 | 5.53 |
> | w/o $w^{sim}$ ($\alpha = 0$) | 0.25 | 0.18 | 0.17 | **0.89** | 9.29 |
> | w/o $w^{per}$ ($\alpha = 1$) | 0.35 | 0.24 | 0.28 | 3.05 | **0.46** |
> | w/o both | 0.40 | 0.68 | 1.24 | 3.80 | 82.7 |
>
> Once again, we appreciate your valuable feedback and hope our response has addressed your concerns.

---

> > ### Comment · Reviewer_V8Sn · 2025-08-07
> >
> > Thanks to the authors for the responses. I will raise my score as my initial concerns have all been addressed, and the authors' responses to other reviews also show more insights and results to strengthen the paper.

---

> > > ### Author Response · Authors · 2025-08-07
> > > **Thank You for Raising the Score**
> > >
> > > Dear reviewer,
> > >
> > > We’re very pleased to hear that our responses effectively addressed your concerns. Thank you sincerely for your feedback and for raising your score. We greatly appreciate your support.

---

> ### Author Response · Authors · 2025-08-07
> **Follow-up on Rebuttal and Feedback**
>
> Dear Reviewer V8Sn,
>
> Thank you again for your initial review. We have carefully addressed all your concerns in our rebuttal, and we hope our responses have effectively clarified the issues you raised.
>
> As the discussion phase is coming to a close, we would greatly appreciate your feedback, whether our responses have resolved your concerns, or if there are further points you'd like to discuss. If you find our replies satisfactory, we would be grateful if you could consider updating your score accordingly.
>
> Best regards,
>
> Authors of Submission #20587

---

### Comment · Area_Chair_tVaa · 2025-08-04

Dear Reviewers,

As you may know, the reviewer-author discussion period for NeurIPS 2025 is currently underway (July 31 – August 6, 11:59pm AoE).
The authors have put in significant time and effort to the rebuttal. Your timely engagement is crucial—not only to acknowledge their responses, but also to help clarify any outstanding issues, ensure a fair and thorough evaluation process, and improve the overall quality of the review cycle.

I kindly ask you to:

- Read the author rebuttal and other reviews (if you have not already).
- Post your response to the author rebuttal as soon as possible.
- Engage in discussion with the authors and other reviewers where appropriate.

Thank you again for your time and contributions to the review process.

Best regards,

Area Chair, NeurIPS 2025

---

### Note · Authors · 2025-08-12

Dear AC and reviewers,

We sincerely thank you for the constructive and insightful feedback. We are pleased that our responses and additional experiments were well received, and that all reviewers agreed their concerns were effectively addressed. We appreciate your willingness to increase the scores in recognition of these improvements.

As a quick note, we thank reviewer **vuKu** for expressing the intention to raise the score. However, we noticed that the interface currently does not reflect the updated score (still shown as 3). If convenient, could you please kindly double-check the rating, and we hope this can be taken into account in the final evaluation.

Once again, we thank all reviewers for your valuable input and contributions to improving our work. We will incorporate the relevant discussions and new results into the revised paper.

Best regards,

Authors of Submission #20587

---

### Decision · Program_Chairs · 2025-09-17

**Decision:**

Accept (poster)

**Comment:**

This paper proposes CenRA, a centralized reward agent framework for multi-task reinforcement learning. By decoupling reward generation from policy optimization, the method enables knowledge transfer through dense shaped rewards, addressing the challenge of sparse rewards and accelerating learning across tasks. Experiments on benchmarks demonstrate improvements over baselines and robustness in transferring to unseen tasks. The reviewers highlighted several strengths: the conceptual clarity of the framework and its empirical validation with extensive baselines. Overall, the consensus is positive. I recommend Accept.